# FinRipple: Aligning Large Language Models with Financial Market for Event Ripple Effect Awareness

## Abstract

Event studies have been fundamental in finance, focusing on analyzing the ripple effects of sudden market events. Accurately predicting these effects is crucial for informed decision-making and effective risk management. However, the dynamic complexity of financial markets and the lack of unified modeling tools make this task challenging. Previous models, constrained by simplistic assumptions and limited scopes, have struggled to address this complexity effectively. In contrast, large language models (LLMs), with their emergent reasoning abilities, offer a promising solution. In this paper, we introduce **FinRipple**, a novel training framework that enables LLMs to align with market behavior and develop the capability to analyze the ripple effects of sudden events. We first construct a time-varying financial knowledge graph (KG) that is both financially meaningful and noise-reduced to accurately represent the market state. These KGs are then integrated into the LLM using adapters as memory modules. Additionally, we align the LLM with market dynamics by integrating **FinRipple** with classic asset pricing theories through a reinforcement learning framework. This market-alignment process collects feedback that enhances the LLM's foundational ability to analyze financial events and explain market anomalies that traditional models fail to address. Our key contributions are as follows: (1) We are the first to define the underexplored task of "event impact prediction". Our framework not only establishes this task but also provides an open-source benchmark, creating a unified evaluation standard for both academia and industry; (2) **FinRipple** complements classic asset pricing models by combining strong theoretical foundations with AI-driven capabilities, offering an enhanced analysis of residuals unexplained by traditional models. We also demonstrate its potential for practical applications such as portfolio management; (3) We conduct a comprehensive analysis to ensure that the results generated by LLMs in our framework are more logically consistent and credible, thus improving the reliability of insights for financial decision-making.

## 1 Introduction

Event studies have been extensively used to determine the impact of corporate announcements and market events on the market value of firms (Sorescu et al., 2017). A well-known recent example underscores the significance of understanding such market reactions: On August 13th, 2024, Starbucks announced that it would replace its CEO with Chipotle CEO Brian Niccol. This announcement led to a remarkable shift in the market, sending Starbucks' stock soaring by 24.5%, marking its best day ever, while Chipotle's stock plummeted by over 10%. Some related companies in Starbucks' supply chain were also affected. For example, Jones Soda Co. saw its stock rise by 9.52%, BRC Inc. gained 6.25%, and Celsius Holdings Inc. increased by 3.81%. This example demonstrates the ripple effect that a single market event can have, not just on the company involved, but on other relevant companies (Ma et al., 2023). Predicting these market ripple effects is crucial for financial decision-making and risk management. Investors and risk managers rely on such insights to anticipate broader company announcements (Boyd et al., 2010; Wu et al., 2015), external news or reviews (Xiong & Bharadwaj, 2013; Gao et al., 2015), or macroeconomic shocks (Chen et al., 2012), allowing them to optimize portfolios, mitigate risks, and act swiftly in volatile conditions (Ding et al., 2015; 2014).

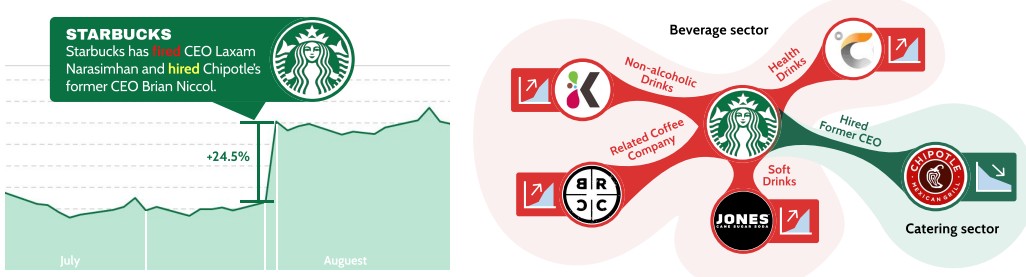

Figure 1: An example of market ripple effects. The announcement of Starbucks's CEO change not only boosted its stock but also positively impacted other related companies in the beverage sector.

However, predicting these ripple effects remains a complex and underexplored challenge because of the intricate, evolving, and interconnected factors at play.

A significant drawback is that financial markets are often much more complex than previously assumed, making it difficult for previous approaches to fully capture the intricate and dynamic nature of market behaviors. Previous research on event studies has mainly followed two main directions: case-by-case analysis and unified modeling based on learning theory. Traditional case-by-case studies typically focus on understanding how market events impact the stock performance of a single company or a group of companies within the same industry. For example, Austin (1993) measured the innovative output of patents within the biotechnology industry. Lepetit et al. (2004) discussed the effects of mergers and acquisitions (M&As) in the banking industry. Ramiah et al. (2013) analyzed how the stock market reacts to the announcement of green policies. While these studies are valuable in assessing direct consequences, they often struggle to capture the ripple effects even across different industries, let alone the complexities of the entire market. Unified modeling based on learning theory studies has mostly used news sentiments of target companies to predict stock price movements and has recognized that considering the information of target companies is insufficient because the stock prices of target companies can be affected by their related companies (Ashtiani & Raahemi, 2023). Recent research has explored the integration of multi-source information (Ma et al., 2023) and employed more advanced embedding models. For instance, several extensions of transformer-based models were utilized by Mishev et al. (2020), demonstrating that combining transformer representations with deep learning classifiers outperforms lexicon-based and statistical models in representing event-driven word embeddings. Although these efforts represent a promising direction for future research, they remain constrained by the limited capacity of the models' capacity and the incompleteness of their frameworks, making it difficult to fully capture the dynamic, time-varying relationships between companies and the broader, evolving financial market. Moreover, focusing solely on text sentiment for classification tasks often results in the loss of critical information. For instance, positive news about one company might negatively impact another company it is associated with. Therefore, up until now, there is an urgent need for a more comprehensive approach to capture the ever-changing market dynamics and explain the complex, interconnected relationships between companies.

Recently, large language models (LLMs) have gained widespread application across numerous fields owing to their powerful reasoning capabilities (Huang & Chang, 2022). LLMs excel at tasks such as structured information extraction (Hao et al., 2024), analogical reasoning (Creswell et al., 2022; Wei et al., 2022b), and question answering, making them particularly well-suited for understanding event-driven ripple effects. Given their potential to model complex interactions, leveraging LLMs for predicting the ripple effects in the financial market is a natural progression. However, directly applying LLMs to financial markets is insufficient. The inherent complexity of these markets, where companies are interconnected in dynamic and often noisy relationships, poses significant challenges (Tang et al., 2022). The relationships between companies are not static; they often react to multiple market events and information (Cheng & Li, 2021). Without considering the timeliness of the market events on which the LLM relies, directly applying these models may result in misleading or inaccurate predictions. To effectively model the ripple effects of market events, it is essential to augment LLMs with time-sensitive, structured knowledge. This ensures that the model captures the latest and most relevant information about the current market state and the evolving relationships between companies.

A feasible solution to this challenge is the integration of a time-varying financial knowledge graph (KG). The financial KG provides a structured, noise-reduced view of the market, offering a clear representation of up-to-date relationships between companies. By continuously updating the KG with the latest events, we can maintain a reliable snapshot of the current market structure (Yang et al., 2023b). This approach allows us to model how companies interact with each other and how those relationships evolve over time, capturing the dynamic nature of financial markets (Cheng et al., 2020). To integrate this knowledge into the LLM, we adopt an adapter-based approach, enabling us to inject the structured information from the KG directly into the LLM without the need to retrain the model from scratch. This not only avoids the potential information loss that could occur with a retrieval-based approach but also provides an easily extendable framework. After training the LLM backbone, new market states can simply be encoded into the adapter for inference. By aligning the LLMs with the financial market and leveraging the power of the KGs, they gain the ability to analyze the ripple effects of events based on the current market structure.

We validate the effectiveness of our framework in asset pricing and portfolio construction, supported by extensive training on real-world data. Additionally, we conduct systematic analyses and case studies to demonstrate that the model's reasoning process in real markets is reliable. The contributions of this work can be summarized as follows:

- We systematically define the "event impact prediction" task and establish an open-source benchmark that provides a unified evaluation standard.

- We introduce **FinRipple**, an easily extensible training paradigm that can transform most LLMs into specialized financial event analysts, enabling them to accurately predict the scope of event impacts.

- We rigorously validate our training framework, **FinRipple**, which augments the LLM with a time-varying KG and aligns it with the financial market. We showcase its strong potential for real-world applications, such as asset pricing and portfolio management. Furthermore, detailed analyses illustrate the model's reasoning pathways, confirming its ability to provide reliable insights into the causal relationships driving market impacts.

## 2 RELATED WORK

### 2.1 EVENT STUDIES IN FINANCE

Event studies have been extensively employed to assess the impact of significant events on asset prices and market behavior (Sorescu et al., 2017). An event can be a firm announcement (e.g., the appointment of a new CMO) or an announcement made by competitors or regulatory bodies that can affect the value of the focal firm (Acquisti et al., 2006). For example, Austin (1993) measured the innovative output of patents within the biotechnology industry; Lepetit et al. (2004) discussed the effects of M&As in the banking industry; and Ramiah et al. (2013) analyzed the stock market reaction to green policy announcements. Although these methods have provided valuable insights, they often struggle to capture the complexity and dynamics inherent in modern financial markets.

Recognizing these limitations, researchers have explored unified modeling approaches based on learning theory, typically utilizing news sentiment analysis to predict stock price movements (Zhang & Skiena, 2010; Pagolu et al., 2016). Recent advancements include the integration of multi-source information (Ma et al., 2023), the employment of more advanced embedding models (Kilimci & Akyokuş, 2019; Mishev et al., 2020), and usage of large language models (LLMs) (Wu et al., 2023; Yang et al., 2023a). Despite these promising developments, existing models struggle to fully capture the dynamic, time-varying relationships between companies and the evolving financial market. Recent efforts on LLMs for financial tasks have aimed to overcome these challenges through multi-agent systems (Yu et al., 2024b;a; Zhang et al., 2024a) and by infusing financial trading knowledge (Zhang et al., 2024b; Li et al., 2023). Considering the structured, dynamic representations provided by knowledge graphs (KGs) (Zhang et al., 2023), **FinRipple** takes an alternative approach by combining LLMs with financial KGs to capture ever-changing market dynamics and explain complex intercompany relationships.

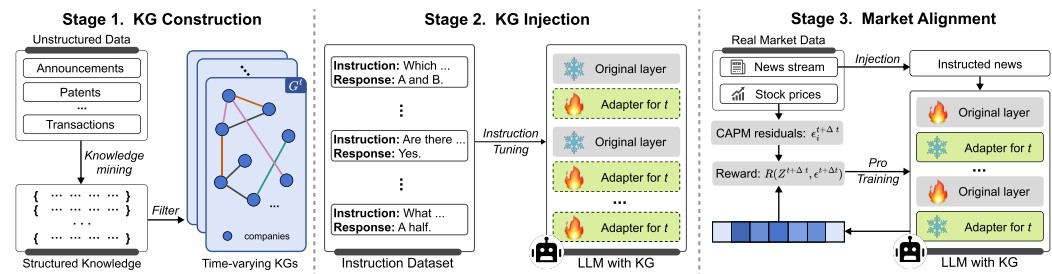

Figure 2: Overview of **FinRipple**. The framework comprises three stages: (1) KG Construction: transforming unstructured data, such as announcements, patents, and transactions, into time-varying KGs that capture company relationships; (2) KG Injection: creating instruction datasets based on these KGs and using them to inject structured knowledge into adapters of an LLM without retraining the original layers; (3) Market Alignment: aligning predictions with real market reaction by using the correlation between the predicted event impact and CAPM residuals as the reward for PPO to optimize model performance. The adapter is frozen, and the analysis ability is parameterized into the original layers of the LLM.

## 2.2 KG Augmented LLM

Through the augmentation of KGs, existing methodologies aim to mitigate hallucinations, enhance reasoning capabilities, and recall specific facts (Chen et al., 2024; Agrawal et al., 2023). Research on using KGs to enhance LLMs can be categorized into two main directions (Wen et al., 2023; Agrawal et al., 2023): 1) integrating KGs into LLM pre-training and 2) injecting KGs into LLM inference. For methods that integrate KGs into LLM pre-training, the common practice involves designing knowledge-aware training objectives by either incorporating KG entities and relations into the training data (Zhang et al., 2019; Sun et al., 2021) or applying KG prediction tasks, such as link prediction, as additional supervision (Yasunaga et al., 2022). These methods directly compress KG knowledge into the parameters of LLMs through supervision. However, creating KGs with trillions of words is challenging, and these methods do not address the fundamental limitations of LLMs regarding flexibility, reliability, and transparency.

Injecting structured symbolic knowledge from KGs into LLM inference aims to enhance contextual understanding, primarily by incorporating them at the input level. Early efforts focused on fusing KG triples into the inputs of LLMs using attention mechanisms (Liu et al., 2020; Sun et al., 2020) or attaching graph encoders to LLM encoders to process KG inputs (Wang et al., 2019). Subsequent work further adopted graph neural networks (GNNs) in parallel with LLMs for joint reasoning (Yasunaga et al., 2021) and added interactions between text tokens and KG entities in the intermediate layers of LLMs (Zhang et al., 2022; Yao et al., 2023).

## 3 METHODOLOGY

### 3.1 PROBLEM FORMULATION

To predict the ripple effects of sudden financial events in dynamic markets, we reformulate this challenge as a structured learning problem. We model the market as a KG $G^t$, defined as:

$$C^t = \{c_1^t, c_2^t, \ldots, c_n^t\}, \quad R^t = \{r_{i,j,k}^t \mid c_i^t, c_j^t \in C^t, k = 1, 2, \ldots, K\}, \quad G^t = \{C^t, R^t\} \quad (1)$$

Here, $n$ is the number of companies at time $t$, $C^t$ is the set of companies in the financial market, and $R^t$ denotes the relationships between them. Each relationship $r_{i,j,k}^t$ specifies an interaction of type $k$ between companies $c_i^t$ and $c_j^t$, with $k$ representing one of $K$ possible relationship types (e.g., supply chain links, mutual fund holdings).

This task can be framed as a more intricate "link prediction problem". To account for sudden events, we introduce a set of events $E^t = \{e_1^t, e_2^t, \ldots, e_m^t\}$, where $m$ is the number of events at time $t$.

This link prediction problem involves expanding the KG to $G^t = \{C^t, E^t, R^t\}$, where the edge set $R^t \subseteq (C^t \times C^t) \cup (C^t \times E^t)$ represents relationships both between companies and between companies and events. Our goal is to predict not just link existence but also the influence of events on companies, formalized as a function $f : C^t \times E^t \to [-I, +I]$, where $f(c_i^t, e_j^t)$ represents the impact of event $e_j^t$ on company $c_i^t$. Positive values of $f$ indicate positive impacts, and higher magnitudes reflect stronger influence. A value of $f(c_i^t, e_j^t)$ equal to zero denotes no measurable impact.

This task uses both the KG and real-time news data as inputs. The KG captures structural relationships between companies, while news data provides event-specific context relevant to financial market dynamics. The output is a matrix $Y^{t+\Delta t} \in \mathbb{R}^{n \times m}$, where each element $Y_{ij}^{t+\Delta t} = f(c_i^t, e_j^t)$ quantifies the predicted impact of event $e_j^t$ on company $c_i^t$. The time shift $\Delta t$ accounts for the finite lag in market reactions to events, as financial markets typically respond to news extremely quickly, with impacts often lasting only 1-2 days (Hafez & Xie, 2016). Our objective is to minimize the following loss function:

$$\min_\theta \sum_{i=1}^n \sum_{j=1}^m dist \left( f_\theta(c_i^t, e_j^t | G^t) - r(c_i^t, e_j^t) \right) \tag{2}$$

where $f_\theta(c_i^t, e_j^t \mid G^t)$ is the predicted influence parameterized by $\theta$, $dist$ is a measurable distance function, and $r(c_i^t, e_j^t)$ represents the true influence. However, directly observing $r(c_i^t, e_j^t)$ from the market is challenging, as a company's daily return may be influenced by various factors. To address this, we deeply integrate this task with classical asset pricing models to filter out multiple influences. Specifically, we use the Capital Asset Pricing Model (CAPM) (Sharpe, 1964) adjusted residuals to approximate $r$. Ideally, these residuals represent the portion of a company's returns that cannot be explained by broader market trends, such as systematic risk factors accounted for in the CAPM model. By focusing on this unexplained component, we attempt to attribute it, at least partially, to the impact of sudden financial events.

## 3.2 The pipeline of FinRipple

In this section, we introduce the implementation details of **FinRipple**. As shown in Figure 2, **FinRipple** starts with the construction of time-varying KGs that incorporate four relationships supported by prior research: leadership networks, mutual fund holdings, patent relationships, and supply chains. The specific data sources and construction process for the KG can be found in Appendix A.2. The next two key steps are KG injection and market alignment, which we will introduce in the following subsections.

### 3.2.1 Knowledge graph Injection

We first convert the time-varying KGs into instruction datasets, primarily composed of three types of questions: retrieval questions, factual judgments, and factual questions, as described in Appendix D.1. We also validate the necessity of including these three types of questions through our experiments, with detailed results provided in Table 8 in the Appendix D.1. For a simplified example, suppose the KG contains a relationship such as "Company A has an upstream supply relationship with Company B." This relationship is transformed into an instruction-response pair as follows: **Instruction:** "Which companies have an upstream supply relationship with Company A?" **Response:** "Company B is an upstream supplier of Company A."

The instruction dataset at a specific time $t$, denoted as $\mathcal{D}^t = \{(x_1^t, y_1^t), \ldots, (x_N^t, y_N^t)\}$, consists of pairs of questions $x_i^t$ and their corresponding responses $y_i^t$, which are generated from the knowledge graph $G^t$. These pairs are designed to effectively teach the model the relationships between companies. When training on $G^t$, an adapter is saved to store the market structure information. Notably, if the model's backbone parameters are modified, the adapter must also be updated accordingly to ensure proper adaptation.

### 3.2.2 Market Alignment

We employ a reinforcement learning framework (Schulman et al., 2017) to fine-tune the LLM backbone while keeping the adapter layers frozen. Before the training process, for each news item, we

retrieve the corresponding KG for the relevant time and inject it into the adapter, enabling the model to adapt to the time-varying market structure. Importantly, each time we fine-tune the backbone of the LLM, the adapter, which stores the information of the KG, is reinitialized and then kept frozen, ensuring compatibility between the updated backbone parameters and the dynamically injected knowledge. The adapter, once frozen, functions as a static feature extractor that represents market features at specific times. Meanwhile, the LLM backbone learns to make predictions consistent with the current market context.

After initializing the market structure information through adapter fine-tuning, we further align the model's predicted impacts with actual market responses by defining a reward function $r_j^{t+\Delta t}$, which assesses the accuracy of the predictions by comparing the deviations between the CAPM residuals and the predicted impacts. The expected return $E(R_i^{t+\Delta t})$ is calculated using the CAPM, while the residual $\epsilon_i$, representing the portion of returns not explained by the CAPM model, captures market-independent influences. If our model's predictions accurately explain these residuals, it indicates the effectiveness of the event-driven impact estimation. The CAPM residual is calculated as follows:

$$E(R_i^{t+\Delta t}) = R_f + \beta_i \left( R_m^{t+\Delta t} - R_f \right), \quad \epsilon_i = R_i^{t+\Delta t} - E(R_i^{t+\Delta t}), \tag{3}$$

where $R_f$ is the risk-free rate (typically the return of short-term government bonds), $\beta_i$ represents the sensitivity of stock $i$ to market returns, and $R_m^{t+\Delta t}$ is the market return at time $t+\Delta t$. The market risk premium is represented by the term $R_m^{t+\Delta t} - R_f$. The stock's expected return is estimated using the following: $R_i^{t+\Delta t} = \frac{P_i^{t+\Delta t} - P_i^t}{P_i^t}, \quad R_m^{t+\Delta t} = \frac{P_m^{t+\Delta t} - P_m^t}{P_m^t}$ where $P_i^{t+\Delta t}$ and $P_m^{t+\Delta t}$ are the prices of stock $i$ and the market index at time $t + \Delta t$, respectively. The coefficient $\beta_i$, indicating stock-market sensitivity, is estimated using ordinary least squares (OLS). The objective function for stock $i$ is given by:

$$\min_{\beta_i} \sum_{t+\Delta t} (\epsilon_i^{t+\Delta t})^2 = \sum_{t+\Delta t} \left( R_i^{t+\Delta t} - \left( R_f + \beta_i(R_m^{t+\Delta t} - R_f) \right) \right)^2 \tag{4}$$

We aggregate the event impact matrix $Y^{t+\Delta t} \in \mathbb{R}^{n \times m}$ to obtain the total impact $Z^{t+\Delta t} \in \mathbb{R}^{1 \times m}$, where $Z_j^{t+\Delta t} = \sum_{i=1}^{n} Y_{ij}^{t+\Delta t}$, representing the cumulative impact of events on company $j$. We then calculate the similarity between $Z^{t+\Delta t}$ and the residual vector $\epsilon^{t+\Delta t} \in \mathbb{R}^{1 \times m}$, defining the reward function $R$ as:

$$R(Z^{t+\Delta t}, \epsilon^{t+\Delta t}) = \frac{Z^{t+\Delta t} \cdot \epsilon^{t+\Delta t}}{\|Z^{t+\Delta t}\| \|\epsilon^{t+\Delta t}\|} + \lambda \frac{\sum_i \min(Z_i^{t+\Delta t}, \epsilon_i^{t+\Delta t})}{\|\epsilon^{t+\Delta t}\|_1} \tag{5}$$

The first term of the above reward function measures how precisely the predicted impacts can explain the CAPM residuals, ensuring the model accurately learns the influence magnitude of specific events. At the same time, the regularization controlled by the hyperparameter $\lambda$ maximizes the recall rate to cover as many relevant impacts as possible. The role of the regularization term is to evaluate the extent to which $Z^{t+\Delta t}$ covers $\epsilon^{t+\Delta t}$ by comparing their values element by element (during training, $\Delta t$ is set to 1.) More training details can be found in Appendix B.

## 4 EXPERIMENT

### 4.1 BASELINES AND EVALUATION METRICS

In this subsection, we provide a brief introduction to the benchmarks and metrics for the asset pricing task only. For further details and information on downstream tasks related to portfolio management, please refer to Appendix E.

**Datasets** We selected 10,000 news articles about S&P 500 companies from January 1, 2020, to June 30, 2022, as the test set, while approximately 110,000 articles from other years were used for training. Detailed statistics on the dataset about news and KGs can be found in Appendix E.

**Baselines**  We adopt several mainstream methods to demonstrate that **FinRipple** offers a powerful solution for this task. The baselines are primarily divided into two categories. The first category tests the analogical reasoning capabilities of foundational LLMs, demonstrating that untrained LLMs lack the ability to effectively analyze event impact. The basic Retrieval-Augmented Generation (RAG) (Lewis et al., 2020) approach utilizes an embedding model to retrieve relevant subgraph information from the KG, enabling LLMs to assess impacts based on this data. Zero-Shot Inference involves providing instructions to the model along with news and concatenated graph information. However, due to the limited window size of LLMs, some graph data may be incomplete. For companies specifically mentioned in the news, a two-hop subgraph is concatenated; otherwise, random graph information is used until the LLM's input window is filled. In-Context Learning (ICL) (Brown et al., 2020) builds upon the Zero-Shot approach by adding an example to aid the LLM in reasoning. The second category primarily includes fine-tuned variations of **FinRipple**, both without and with market alignment. It emphasizes that even if the LLM effectively absorbs the graph information, without aligning with market dynamics, the model still lacks the ability to effectively analyze the impact of events.

**Evaluation metrics**  To evaluate the effectiveness of **FinRipple** in analyzing financial market shocks, we designed an evaluation framework focusing on three metrics: (1) the explanatory power on the mean of the residuals, (2) the explanatory power on the variance of the residuals, and (3) the refusal-to-answer rate. The residuals, derived from a CAPM regression of stock returns against market returns, represent the portion of returns unexplained by market factors. We use these residuals to assess whether predicted event impacts significantly explain the variance in returns through regression analysis and ANOVA, with $p$-values indicating statistical significance. Additionally, the refusal-to-answer rate evaluates the robustness of LLMs in generating meaningful responses in complex, event-driven contexts.

## 4.2 MAIN RESULT

| Model | RAG | | | Zero-Shot | | | ICL | | | FinRipple/w-o alignment | | | FinRipple | | |
|---|---|---|---|---|---|---|---|---|---|---|---|---|---|---|---|
| | Coef. | p-value | $R^2$ | Coef. | p-value | $R^2$ | Coef. | p-value | $R^2$ | Coef. | p-value | $R^2$ | Coef. | p-value | $R^2$ |
| llama2-7b-chat | 0.012 | 0.452 | 0.009 | 0.031 | 0.601 | 0.012 | 0.042 | 0.503 | 0.018 | 0.047 | 0.510 | 0.019 | 0.150* | 0.030 | 0.083 |
| llama2-13b-chat | 0.103 | 0.305 | 0.054 | 0.079 | 0.349 | 0.039 | 0.098 | 0.281 | 0.061 | 0.102 | 0.287 | 0.058 | 0.242** | 0.009 | 0.193 |
| llama3-8b-instruct | 0.091 | 0.318 | 0.047 | 0.072 | 0.402 | 0.037 | 0.107 | 0.254 | 0.058 | 0.110 | 0.249 | 0.060 | 0.278** | 0.004 | 0.251 |
| vicuna-7b-chat | 0.118 | 0.247 | 0.063 | 0.102 | 0.298 | 0.052 | 0.129 | 0.198 | 0.081 | 0.125 | 0.205 | 0.074 | 0.330*** | 0.001 | 0.310 |
| vicuna-13b-chat | 0.248* | 0.032 | 0.248 | 0.148 | 0.149 | 0.082 | 0.176 | 0.098 | 0.102 | 0.171* | 0.040 | 0.108 | 0.395*** | 0.000 | 0.340 |
| Phi-3.5-mini-instruct | 0.082 | 0.395 | 0.032 | 0.065 | 0.498 | 0.019 | 0.094 | 0.347 | 0.052 | 0.096 | 0.340 | 0.045 | 0.245** | 0.006 | 0.155 |
| gemma-2-9b-it | 0.097 | 0.298 | 0.048 | 0.083 | 0.354 | 0.038 | 0.112 | 0.245 | 0.063 | 0.109 | 0.252 | 0.061 | 0.290*** | 0.001 | 0.215 |
| GPT 3.5 | 0.083 | 0.398 | 0.028 | 0.062 | 0.051 | 0.075 | 0.056** | 0.004 | 0.112 | / | / | / | / | / | / |
| GPT o1-preview | 0.152 | 0.342 | 0.047 | 0.119 | 0.392 | 0.056 | 0.192 | 0.229 | 0.082 | / | / | / | / | / | / |
| GPT 4o-mini | 0.124 | 0.312 | 0.042 | 0.312* | 0.013 | 0.035 | 0.104 | 0.879 | 0.103 | / | / | / | / | / | / |

Table 1: Comparison of baselines and **FinRipple** on LLMs. This table focuses on the explanatory power on the value of the CAPM residuals. The significance levels are indicated as follows: * $p < 0.05$, ** $p < 0.01$, *** $p < 0.001$. Note that cells containing a slash (/) indicate that the model does not have open-sourced weights available.

As shown in Table 1, both open-source and closed-source LLMs face significant challenges in analyzing the impact of financial market events without domain-specific training. Despite their strong capabilities, such as those seen in the GPT series, these models exhibit limited explanatory power in complex, event-driven scenarios, as indicated by their relatively low $R^2$ values, which measure the proportion of variance in residuals explained. The RAG method, in particular, heavily relies on the embedding model's ability to extract event-relevant subgraphs. The volatility in RAG's performance underscores its limitations; for instance, although vicuna-13b-chat achieves an $R^2$ of 0.248 with a p-value of 0.032, this result reveals inherent bottlenecks in its capabilities. Similarly, ICL, which attempts to improve performance by including examples within the input context, offers very limited enhancement. For example, the llama2-13b-chat model achieves an $R^2$ of 0.061 under ICL, which represents only a marginal improvement over Zero-Shot performance ($R^2$ = 0.039), indicating a minimal impact on its reasoning over event-driven data. However, models utilizing knowledge injection through **FinRipple** without alignment exhibit modest gains by incorporating broader market information. In stark contrast, models that undergo domain-specific fine-tuning with **FinRipple** show significant performance improvements. For example, the llama2-13b-chat model's $R^2$ score increases to 0.193 after fine-tuning, demonstrating an enhanced ability to generalize and effectively

capture the impacts of market events. Additionally, vicuna-7b-chat experiences a substantial improvement, with its $R^2$ increasing from 0.072 under ICL to 0.310 following **FinRipple** alignment. This highlights the crucial role of aligning LLMs with market dynamics, irrespective of the original model size or capabilities.

Furthermore, a notable gap between small and large models is observed. For example, vicuna-7b-chat scores an $R^2$ of 0.340 after **FinRipple** alignment, which demonstrates that larger models possess an inherent capacity to learn complex market dynamics.

| Model | RAG | | | Zero-Shot | | | ICL | | | FinRipple/w-o alignment | | | FinRipple | | |
|---|---|---|---|---|---|---|---|---|---|---|---|---|---|---|---|
| | ANOVA-F | ANOVA-p | ES | ANOVA-F | ANOVA-p | ES | ANOVA-F | ANOVA-p | ES | ANOVA-F | ANOVA-p | ES | ANOVA-F | ANOVA-p | ES |
| llama2-7b-chat | 1.624 | 0.231 | 0.089 | 1.304 | 0.274 | 0.068 | 2.392 | 0.097 | 0.108 | 2.565 | 0.082 | 0.092 | 3.123* | 0.033 | 0.142 |
| llama2-13b-chat | 2.175 | 0.139 | 0.102 | 1.782 | 0.188 | 0.082 | 2.634 | 0.075 | 0.117 | 3.052* | 0.051 | 0.105 | 4.103** | 0.012 | 0.198 |
| llama3-8b-instruct | 1.210 | 0.324 | 0.085 | 2.221 | 0.141 | 0.099 | 2.452 | 0.088 | 0.112 | 2.835 | 0.069 | 0.101 | 4.110** | 0.010 | 0.203 |
| vicuna-7b-chat | 0.910 | 0.452 | 0.071 | 1.512 | 0.248 | 0.074 | 2.731 | 0.060 | 0.115 | 2.672 | 0.074 | 0.097 | 3.832* | 0.019 | 0.341 |
| vicuna-13b-chat | 2.703 | 0.112 | 0.115 | 2.910* | 0.058 | 0.110 | 3.001* | 0.052 | 0.125 | 3.932** | 0.031 | 0.119 | 5.231*** | 0.003 | 0.287 |
| Phi-3.5-mini-instruct | 1.563 | 0.257 | 0.097 | 2.334 | 0.126 | 0.104 | 2.815 | 0.062 | 0.118 | 3.014* | 0.048 | 0.110 | 4.315** | 0.009 | 0.215 |
| gemma-2-9b-it | 2.443 | 0.128 | 0.109 | 1.905 | 0.172 | 0.091 | 2.447 | 0.089 | 0.095 | 3.122* | 0.039 | 0.108 | 4.012** | 0.014 | 0.159 |
| GPT 3.5 | 1.375 | 0.301 | 0.090 | 1.645 | 0.223 | 0.088 | 2.087 | 0.129 | 0.105 | / | / | / | / | / | / |
| GPT 4.0-preview | 0.812 | 0.443 | 0.067 | 2.112 | 0.145 | 0.100 | 2.372 | 0.098 | 0.117 | / | / | / | / | / | / |
| GPT 4o-mini | 2.153 | 0.144 | 0.099 | 2.875* | 0.059 | 0.108 | 3.245 | 0.061 | 0.145 | / | / | / | / | / | / |

Table 2: Comparison of baselines and **FinRipple** on various models using ANOVA analysis. ANOVA-F represents the F-value from the ANOVA test, indicating the ratio of systematic variance to error variance. ANOVA-p represents the p-value for statistical significance, with * indicating $p < 0.05$, ** indicating $p < 0.01$, and *** indicating $p < 0.001$. Eta Squared (ES) represents the correlation ratio, which indicates the proportion of variance explained by the model. Cells with a slash (/) indicate that the model cannot be fine-tuned using **FinRipple** due to unavailable open-source weights.

In line with our experience, the refusal-to-answer rate largely depends on the model's instruction-following capability. As shown in Table 3, Zero-Shot methods generally perform poorly across all models, with high variability in refusal rates, such as 0.41 ± 0.16 for llama2-7b-chat and 0.48 ± 0.21 for Phi-3.5-mini-instruct. This indicates that Zero-Shot methods have limited ability to comprehend instructions for complex financial tasks and are highly sensitive to decoding parameters. In contrast, closed-source models like GPT 4.0-preview and GPT 4o-mini demonstrate significantly lower refusal rates, at 0.14 and 0.12 respectively, reflecting their stronger instruction-following capabilities.

| Model | Zero-Shot | ICL | FinRipple |
|---|---|---|---|
| llama2-7b-chat | 0.41 ± 0.16 | 0.25 ± 0.09 | 0.21 ± 0.11 |
| llama2-13b-chat | 0.36 ± 0.18 | 0.13 ± 0.08 | 0.15 ± 0.09 |
| llama3-8b-instruct | 0.45 ± 0.19 | 0.11 ± 0.07 | 0.14 ± 0.08 |
| vicuna-7b-chat | 0.39 ± 0.14 | 0.22 ± 0.10 | 0.23 ± 0.05 |
| vicuna-13b-chat | 0.34 ± 0.15 | 0.13 ± 0.02 | 0.10 ± 0.04 |
| Phi-3.5-mini-instruct | 0.48 ± 0.21 | 0.31 ± 0.12 | 0.26 ± 0.09 |
| gemma-2-9b-it | 0.38 ± 0.17 | 0.23 ± 0.08 | 0.18 ± 0.06 |
| GPT 3.5 | 0.32 | 0.18 | / |
| GPT 4.0-preview | 0.14 | 0.10 | / |
| GPT 4o-mini | 0.12 | 0.09 | / |

Table 3: Refusal-to-Answer Rate Comparison. The fluctuating values indicate the range of variation under different temperature settings. This experiment is conducted on our benchmark, where refusal-to-answer samples are those that could not be post-processed into valid outputs.

The effectiveness of **FinRipple** also varies depending on the model, with a significant reduction in refusal rate for models like vicuna-13b-chat (0.10 ± 0.04). This suggests that effective instruction design plays a crucial role in achieving better model alignment and performance. Furthermore, we recognize that instruction-following ability is a key factor in FinRipple's effectiveness. This means that the stronger the base LLM, the greater the effectiveness it can achieve once aligned with the financial market.

## 4.3 PORTOFOLIO MANAGEMENT

To further demonstrate the effectiveness of **FinRipple**, we implement a simple intraday trading strategy based on the event impact prediction. The strategy selects stocks that exhibit the highest positive predicted event-driven impacts and creates a daily portfolio that rebalances at the end of each trading day. Specifically, the steps are as follows:

1. Each morning, based on the predicted impact results, we rank all stocks in our universe by the magnitude of their predicted impact.

2. The top 10% of stocks with the highest predicted positive impact are selected for a long position, while the bottom 10% with the highest predicted negative impact are shorted.

3. At the end of the day, the portfolio is rebalanced, and the next day's selection is based on new predictions.

In accordance with previous portfolio management studies (Xu et al., 2024), we selected several benchmarks, including Equal Weighting, Volatility Weighting, the Markowitz Model, and Min-Variance Weighting. Furthermore, we employed multiple evaluation metrics, such as daily return ($R_d$), sharpe ratio ($S_a$), and maximum drawdown (MDD), as presented in Table 4. To prevent data contamination, the backtest period was set from January 2020 to June 2022, ensuring the reliability of the results. A detailed introduction to portfolio strategies and their evaluation can be obtained in Appendix E. The results clearly demonstrate that accurately predicting the range of impacts from

| Benchmark | Daily Return ($R_d \times 10^{-1}$) | Sharpe Ratio ($S_a$) | Maximum Drawdown (MDD) | Win Rate |
|---|---|---|---|---|
| Equal Weighting | 0.034 | 0.882 | -0.351 | 0.582 |
| Volatility Weighting | 0.041 | 1.021 | -0.312 | 0.643 |
| Markowitz Model | 0.029 | 0.954 | -0.292 | 0.613 |
| Min-Variance Weighting | 0.028 | 0.821 | -0.401 | 0.552 |
| FinRipple | 0.052 | 1.153 | -0.283 | 0.685 |

Table 4: Summary of backtest results for different portfolio management strategies on S&P 500 constituent stocks (January 2020 to June 2022). Note that the daily return is presented with a factor of $10^{-1}$ for better readability.

financial market events can significantly mitigate portfolio risks. The strategy based on **FinRipple** outperforms other benchmarks in key metrics, including daily return, Sharpe ratio, and maximum drawdown, achieving a daily return of $0.052$, a Sharpe ratio of $1.153$, and a maximum drawdown of $-0.283$. In contrast, strategies like Equal Weighting and Min-Variance Weighting exhibit higher maximum drawdowns, indicating greater vulnerability to market shocks when lacking precise impact predictions. Overall, accurate event impact forecasting plays a crucial role in enhancing risk control and improving investment outcomes.

## 4.4 ANALYSIS

### 4.4.1 KNOWLEDGE INJECT ANALYSIS

In this subsection, we first analyze the necessity of knowledge injection. When effectively injecting KGs into LLMs, optimizing the model's understanding of market structures is paramount. One strategy involves using a preprocessing module to filter potential subgraphs as inputs. The simplest approach is to traverse one-hop and two-hop subgraphs related to a target company. While this method may be applicable in some contexts, it fails to capture the market's dynamic complexity, particularly in scenarios where events do not specifically target individual companies, such as those affecting entire supply chains. Another strategy is to leverage RAG, which heavily relies on the

> **Example of a news event not targeting a specific company:**
>
> In August 2021, the Biden administration announced a plan to ***invest \$7.3 billion in the construction of electric vehicle (EV) charging infrastructure.*** This initiative aims to establish 50,000 public charging stations across the United States by 2030, supporting the widespread adoption of electric vehicles. This effort is part of a broader strategy to promote clean energy and reduce carbon emissions, ultimately creating a more environmentally friendly transportation system.

Figure 3: An example where subgraph search is not applicable. As shown in the figure, this news event impacts the entire electric vehicle charging infrastructure industry rather than targeting a specific company.

performance of embedding models designed to recall companies that are "semantically similar" to

specific queries. However, these embedding models often overlook the deeper market relationships associated with specific events when filtering for potentially impacted companies. This dependency can lead to significant misjudgments or biases in the model's event impact predictions.

In contrast, the parameterization approach, which transforms KGs into adjustable parameters, provides a more comprehensive reflection of market trends and their complex interrelationships. This method enables dynamic adjustment and optimization of parameters during training, allowing the model to better capture the nonlinear dynamics of the market. By employing time-varying adapters, the model's adaptability to changes in market structure is enhanced, improving its responsiveness and predictive accuracy regarding market dynamic. For news events that focus on a specific central company, as Figure 4 shows, RAG primarily retrieves based on semantic similarity, which often leads to a low recall rate when dealing with larger graphs. This limitation also affects first- and second-degree nodes, reducing the effectiveness of the retrieval process. Subgraph retrieval without alignment may select a larger number of relevant companies, but it often lacks the necessary logical structure to make meaningful predictions. **FinRipple**, by contrast, effectively captures not only the relationships among entities but also the logical pathways of impact from the central company, offering a more coherent and precise prediction of event impact. The clear propagation routes observed in **FinRipple** highlight its ability to model the cascading effects of an event through the network, accurately representing both direct and indirect influences.

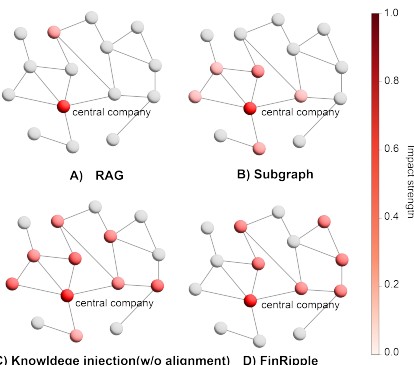

Figure 4: This diagram compares candidate companies identified by **FinRipple** with those identified by other methods. Due to the complexity of the full network, only selected nodes in the examples are shown for illustration purposes.

### 4.4.2 CASE STUDY

Recollection

In January 2020, **MGM** Resorts International sold the **MGM Grand and Mandalay Bay** to a joint venture including **Blackstone Group Inc.** as part of **MGM**'s "asset-light" strategy to divest real estate and focus on sports betting and entertainment.

In 2017, **Hilton Worldwide (HLT)** spun off its real estate assets into **Park Hotels & Resorts (PK)** REIT to focus on hotel management and brand services, simplifying its structure and reducing heavy assets.

Analysis

**Vanguard Group** holds shares in **MGM**. If **MGM** divests its real estate assets, **Vanguard** may adjust investments in similar sectors, potentially affecting **Simon Property Group** holding comparable real estate assets.

**Vanguard Group** holds shares in both **MGM** and **Cisco Systems**. **MGM**'s asset divestiture may reduce demand for network equipment, impacting **Cisco Systems'** revenue and **Vanguard**'s returns on its Cisco holdings.

**Vanguard Group** holds shares in **United Technologies**. **MGM**'s asset divestiture may affect **United Technologies'** performance and **Vanguard**'s investment returns.

Inference

MGM → Vanguard Group → Simon Property Group ...
MGM → Cisco Systems → Vanguard Group ...
MGM → United Technologies → Vanguard Group ...

Figure 5: Using the CoT technique to analyze the reasoning process of vicuna-13b-chat. The model is aligned by **FinRipple**.

We believe that the logical reasoning capability of LLMs lies in their ability to establish connections with previously acquired knowledge or patterns. Therefore, in the inference process, we employ a straightforward Chain-of-Thought (CoT) (Wei et al., 2022a) approach to capture the intricate reasoning pathways, ultimately leading to the refined outcomes of **FinRipple**, as illustrated in the Figure 5. We can clearly observe that the inference process of the LLM, after being aligned with the financial market, is divided into three distinct steps: the first step involves establishing connections with past news, the second step focuses on analysis, and the third step derives the impact pathways. It is worth noting that not all news articles can directly establish connections with past knowledge. News that has undergone pre-training or supervised fine-tuning (SFT) is often more likely to be fully recalled and integrated into reasoning processes.

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

# A    DATASETS DETAILS

Data preparation is critical in ensuring the quality and relevance of the input information for our model. This phase is bifurcated into two primary components: the collection of news events and the construction of the time-varying financial KG.

## A.1    NEWS COLLECTION AND PROCESSING:

The origin 792,684 news articles are sourced from Dow Jones News Services and the Wall Street Journal, and stored as structured XML files. The structured dataset comprises eight variables, including {Publication_datetime, Publisher_name, Region_code, Company_code, Title, Body, Word_count, Action}. Detailed descriptions of these variables are provided in Table 5. Using the 'Company_code' variable, we filtered out 129,753 news articles about individual S&P 500 firms, covering the period from March 8, 2001, to October 30, 2023. After removing the irrelevant variables, the remaining eight variables and their descriptions are detailed in Table 5. Figure 6 (A) illustrates the distribution of news articles over time. Notably, only 2 articles were recorded in 2001, while the highest number of articles, 16,103, was collected in 2012. The analysis of word counts reveals that the average number of words per news article is 5,443.85, with the maximum word count reaching 77,086 and the minimum at 23 words. This variation indicates a wide range of article lengths, from brief news briefs to extensive, in-depth reports. Figure 6 (B) presents the top ten companies with the highest number of news articles in the dataset. This ranking highlights the companies that receive the most media attention, which may be attributed to their market influence, recent activities, or significant corporate actions. We further analyzed the properties of daily news based on the 'Action' variable, as shown in Figure 6 (C). 63.94% of the news articles pertain to organizational adjustments, which include changes in the company's business strategy, personnel, or departmental structures. 36.06% of the news articles involve new initiatives, such as the establishment of new companies, launching new projects or services, hiring new executives, and introducing new product lines, etc.

| Variable | Description |
|---|---|
| Publication_datetime | Date and time of news article publication. It records the exact date and time when the news article was officially published. |
| Publisher_name | Name of the news publisher. It indicates the media outlet or organization that published the news article. |
| Region_code | Geographical region code. It specifies the geographic location relevant to the company's operational area. |
| Company_code | Unique identifier or code for the relevant company. A unique code that identifies the company mentioned in the news. |
| Title | Title of news article. A brief headline that summarizes the main topic or event described in the news article. |
| Body | The detailed news content. |
| Word_count | Number of total word count in the body of the news article. |
| Action | Type of corporate action mentioned in the news. Its value can be 'rep' or 'add'. |

Table 5: The variables in the collected news articles dataset.

## A.2    KNOWLEDGE GRAPH CONSTRUCTION:

We constructed comprehensive financial KGs aimed at capturing the multifaceted interrelationships between companies and their potential impacts on profitability. Each company is represented as a node, while the interrelationships between companies constitute the edges of the KGs. To achieve this, we integrate various types of relationships derived from multiple data sources, ensuring a rich and nuanced representation of corporate interactions.

- **Technical Relevance Relationships.** We collect detailed and comprehensive information on firms' patents, including their corresponding Cooperative Patent Classification (CPC) codes, from the USPTO (United States Patent and Trademark Office) database to ensure a

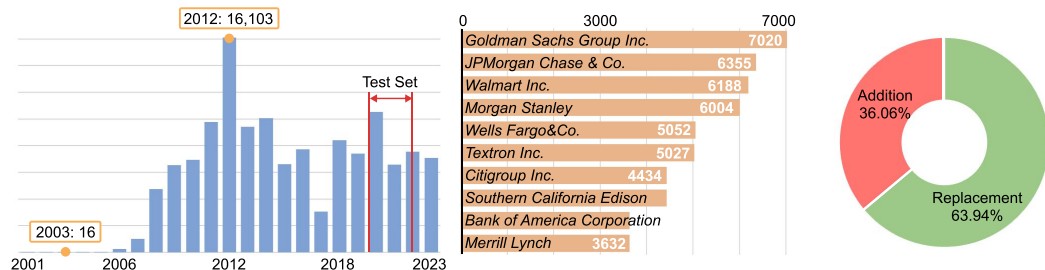

Figure 6: The statistics results of our collected news articles. (A) demonstrates the temporal distribution of news articles, (B) displays the company rankings with the top ten news counts, and (C) shows the properties of different corporate actions.

robust foundation for analyzing technical relevance and relationships between companies. Following the methodology outlined in Lee et al. (2019), we calculate pairwise technical closeness between two firms by measuring the correlation of CPC code distribution across their portfolios. In this kind of KGs, an edge between two companies reflects their patent-based technical similarity. The strength of the edge is proportional to the degree of technical similarity, capturing the depth of their technological connections.

- **Supply Chain Relationships.** Information on firms' supply chains is extracted from the Compustat-Capital IQ database. In this kind of KGs, nodes represent companies, and edges indicate input-output relationships between companies. The strength of an edge is determined by the financial value of transactions between companies, providing a weighted representation of the intensity of their supply chain interactions.

- **Shared Leadership Relationships.** We obtain detailed information on firms' top leaders from the Boardex database. This data highlights interconnections between companies through shared executive affiliations. In this kind of KGs, edges denote the number of directors who simultaneously serve on the boards of two companies. This construction approach quantifies the degree of overlap in leadership structures, capturing the corporate governance ties between firms.

- **Mutual Fund Holding Relationships.** Data on mutual fund holdings of the listed U.S. firms is sourced from the Thomson/Refintiv database. Utilizing this information, we construct the holding-based KGs where an edge between two companies signifies that they are held by the same mutual fund. This relationship reflects the shared ownership structures and potential investment linkages among firms.

By extracting different types of relationships from these diverse data sources, we are able to construct a KG reflecting various dimensions of corporate interactions. In the KG, each company and event is represented as a node, while the interrelationships between companies (such as collaborations or competitions) and the impact of events on companies constitute the edges of the graph.

In the process of constructing the KG, we pay special attention to associations supported by empirical financial research, such as future technology linkages evidenced by patent data and upstream-downstream enterprise relationships. This focus ensures that the KG not only documents the static relationships but also delves deeply into how these relationships influence company performance under varying market conditions and in response to specific events. The resulting KG provides a comprehensive understanding of the interactions among S&P 500 companies and offers the framework a robust and comprehensive understanding foundation.

Our KG dataset is divided into training and testing sets. The training set covers the period from March 2001 to December 2019 (226 months), and the testing set encompasses the period from January 2020 to June 2022 (30 months). Table 6 presents detailed statistics for both the training and testing KGs. It includes the number of contained graphs, the average number of nodes per graph, the average number of edges per graph, and the distribution of relationship multiplicities between nodes.

| | Graphs | Avg. Nodes per Graph | Avg. Edges per Graph | Single Relationship (%) | Dual Relationships (%) | Triple Relationships (%) |
|---|---|---|---|---|---|---|
| Training set | 226 | 6621.6018 | 13,844,186 | 92.7923 | 7.1956 | 0.0104 |
| Testing set | 30 | 6452.1667 | 14,228,088 | 95.0923 | 4.9007 | 0.0053 |

Table 6: KG Data Statistics

## B  FINRIPPLE DETAILS

### B.1  THE DETAILED PIPELINE OF FINRIPPLE

---

**Algorithm 1** Training Pipeline of FinRipple

---

**Training Process:**
**Input:** KG s $G^t = \{G^1, \ldots, G^n\}$, News data $N^t = \{N^1, \ldots, N^m\}$, Pretrained LLM backbone $f_\theta$, Adapters $g_\phi$
**Output:** Updated LLM backbone parameters $\theta^*$

---

1: **for** each time step $t$ **do**
2:     Initialize an empty set $I = \{\}$, collect the KG $G^t = \{C^t, R^t\}$ and news data $N^t = \{n_1^t, \ldots, n_m^t\}$.
3:     **for** each article $n_j^t \in N^t$ **do**
4:         Inject the corresponding KG $G^t$ into the adapter $g_\phi$:

$$g_\phi^t \leftarrow g_\phi(G^t), f_\theta^\phi = g_\phi^t + f_\theta$$

5:         Inference the impact $Y_{ij}^{t+\Delta t}$ based on $n_j^t$:

$$Y_{ij}^{t+\Delta t} \leftarrow f_\theta^\phi(n_j^t), I \leftarrow I \cup Y_{ij}^t$$

6:         Compute the CAPM residuals:

$$\epsilon_i^{t+\Delta t} = R_i^{t+\Delta t} - E(R_i^{t+\Delta t}), E(R_i^{t+\Delta t}) = R_f + \beta_i(R_m^{t+\Delta t} - R_f)$$

7:         Calculate the reward at time $t$:

$$R(Z^{t+\Delta t}, \epsilon^{t+\Delta t}) = \frac{Z^{t+\Delta t} \cdot \epsilon^{t+\Delta t}}{\|Z^{t+\Delta t}\|\|\epsilon^{t+\Delta t}\|} + \lambda \frac{\sum_i \min(Z_i^{t+\Delta t}, \epsilon_i^{t+\Delta t})}{\|\epsilon^{t+\Delta t}\|_1} \quad \text{where } Z_j^{t+\Delta t} = \sum_{i=1}^n Y_{ij}^{t+\Delta t}$$

8:     **end for**
9:     Update $\theta$ based on accumulated rewards.

$$\theta \leftarrow \theta + \alpha \mathbb{E}_t \left[ \nabla_\theta \log f_\theta^\phi(a_t|n_j^t) \frac{f_\theta^\phi(a_t|n_j^t)}{f_{\theta_{\text{old}}}^\phi(a_t|n_j^t)} \hat{A}_t \right] \quad \text{where } \hat{A}_t = R^t - V(n_j^t)$$

10: **end for**

---

**Inference Process:**
**Input:** new event $e_{new}$ and the corresponding KG $G^{t_{new}}$.

---

1: Inject $G^{t_{new}}$ into the frozen adapter $g_\phi$:

$$g_\phi \leftarrow g_\phi(G^{t_{new}})$$

2: Use the fine-tuned LLM backbone $f_{\theta^*}$ to predict the impact of the new event:
    $Y^t = f_{\theta^*}(G^{t_{new}}, e_{new})$ where $Y^t$ represents the predicted impact of $e_{new}$ on the companies $C^t$.
3: Output the predicted impact matrix $Y^t$.

---

## B.2 THE PROMPTS USED IN FINRIPPLE

The following is a detailed prompt designed in **FinRipple** to guide the LLM for financial event analysis. The LLM is instructed to evaluate the impact of news on companies and provide a structured output. The news report will be placed in the "[INSERT MARKET NEWS REPORT]" section. The LLM is expected to determine the affected companies, classify the impact type and assign an impact score from -10 to +10. A high positive or negative score indicates the strength of the potential market effect. The output should include specific company names, detailed descriptions, and adhere strictly to the given format for consistency and clarity. An example is provided within the prompt to illustrate the expected response.

```
Instruction:

You are a financial event analyst focused on analyzing the potential
impacts of news reports on the market. Based on the given news content
and current market structure, evaluate and output the affected companies
, the type of impact (positive, negative, or neutral), and a score
representing the strength of the impact (ranging from -10 to +10,
where -10 indicates a very negative impact, and +10 indicates a very
positive impact). Provide pecific company names and event descriptions
for clarity and utility. Here is an example.

Input Example:

 "Company A announces a partnership with Company B to jointly develop new
 technology, expected to significantly enhance production efficiency
 and increase market share."

Output Format Example:

{
  "impact_analysis": {
    "affected_companies": [
      {
        "name": "Company A",
        "impact_type": "positive",
        "impact_score": 8
      },
      {
        "name": "Company B",
        "impact_type": "positive",
        "impact_score": 6
      }
    ],
    "analysis": "The partnership between Company A and Company B is
    expected to enhance their technological capabilities and market
    competitiveness, likely increasing their revenues and stock prices.
  }
}

Input (you need to analyze):

[INSERT MARKET NEWS REPORT]

Provide your result, strictly following the output format in the
example, without any additional output.
```

# C   ASSET PRICING MODELS

Asset pricing models are essential tools in finance for understanding the relationship between risk and expected return. This appendix briefly introduces three prominent models: CAPM, Fama-French Three-Factor Model (Fama3), and Fama-French Five-Factor Model (Fama5).

## C.1   CAPITAL ASSET PRICING MODEL

The CAPM describes the relationship between systematic risk and expected return. The expected return of an asset is proportional to its beta, which measures the sensitivity of the asset's returns to market returns. The formula for CAPM is:

$$E(R_i) = R_f + \beta_i \left( E(R_m) - R_f \right) \tag{6}$$

where $E(R_i)$ represents the expected return of the asset, $R_f$ is the risk-free rate, $\beta_i$ is the asset's beta that measures its sensitivity to market movements, and $E(R_m)$ is the expected return of the market.

## C.2   FAMA-FRENCH THREE-FACTOR MODEL

The Fama3 expands upon CAPM by including two additional factors: size and value. The size premium, denoted as Small Minus Big (SMB), captures the excess return of small-cap stocks over large-cap stocks, while the value premium, denoted as High Minus Low (HML), captures the excess return of high book-to-market stocks over low book-to-market stocks. The model is represented as:

$$E(R_i) = R_f + \beta_i \left( E(R_m) - R_f \right) + s \times \text{SMB} + h \times \text{HML} \tag{7}$$

where $s$ and $h$ represent the sensitivities of the asset's returns to the SMB and HML factors, respectively.

## C.3   FAMA-FRENCH FIVE-FACTOR MODEL

The Fama5 extends Fama3 by adding two more factors: profitability and investment. The profitability premium, denoted as Robust Minus Weak (RMW), captures the excess return of firms with high profitability over those with low profitability. The investment premium, denoted as Conservative Minus Aggressive (CMA), captures the excess return of firms with conservative investment policies over those with aggressive policies. The updated model is:

$$E(R_i) = R_f + \beta_i \left( E(R_m) - R_f \right) + s \times \text{SMB} + h \times \text{HML} + r \times \text{RMW} + c \times \text{CMA} \tag{8}$$

where $r$ and $c$ represent the sensitivities to the RMW and CMA factors, respectively.

## C.4   RESIDUALS AND MARKET ANOMALIES

Residuals of these models represent the portion of an asset's return not captured by the included risk factors. By analyzing residuals, investors can identify abnormal returns that the models fail to explain. These anomalies often arise due to market inefficiencies, information asymmetries, or other idiosyncratic risks not accounted for by the systematic factors in the models. Understanding residuals helps investors gain insights into potential mispricing and hidden variables in the market, revealing opportunities or risks that standard models overlook.

# D OTHER EXPERIMENTAL RESULTS

## D.1 THE ACCURACY OF KG INJECTION

| Problem Classification | Typical Questions |
|---|---|
| Retrieval Questions | "Which companies have a common CEO relationship with {}?" "Which companies have an upstream-downstream relationship with {}?" "Which companies have multiple relationships with {}?" "Which companies have one relationship with {}?" "Which companies have one relationship with {}?" |
| Factual Judgments | "Are there supply chain upstream and downstream transactions between {} and {}?" "Are the companies {} and {} held by the same fund?" "Are the companies {} and {} held by the same fund?" |
| Factual Questions | "What is the relationship between {} and {}?" "What is the technical similarity between {} and {}?" "What is the technical similarity score between {} and {}?" |

Table 7: The three classes of instruction questions generated from KGs.

| Model | All | w/o RQ | w/o FJ | w/o FQ |
|---|---|---|---|---|
| Gemma-2b-it | 84.6% | 38.5% | 15.4% | 30.8% |
| Gemma-7b-it | 69.2% | 30.8% | 46.2% | 46.2% |
| Llama-13b-chat | 61.5% | 7.7% | 15.4% | 23.1% |

Table 8: Ablation study results for the three classes of questions: Retrieval Questions (RQ), Factual Judgments (FJ) and Factual Questions (FQ). The above results are averaged over five shuffles of the subgraph.

We used a random subgraph of 100 nodes for training, with an 8:2 split between the training and testing datasets. The results indicate that all three types of questions are beneficial. Note that some questions may not be answered correctly if the information needed is not fully covered by the training set. If all information is covered, our tests show that the adapter's memory accuracy reaches approximately 90%. We constructed three types of questions by traversing the KG , as shown in Table 7. The first category, Retrieval Questions, focuses on identifying specific relationships between companies, such as shared CEOs or upstream-downstream connections. The second category, Factual Judgments, is used to determine whether certain relationships exist, such as common fund holdings or supply chain transactions. Finally, the third category, Factual Questions, aims to explore the details of relationships between entities, such as the nature of technical similarities or similarity scores.

## D.2 EVALIDATION ON OTHER ASSET PRICING MODELS

In this subsection, we also evaluate **FinRipple**'s ability to explain the residuals of other models including Fama3 and Fama5. Based on our experimental findings, as shown in Table 9 and Table 10, we observe that the explanatory difficulty of Fama3 and Fama5 residuals gradually decreases. This reduction is primarily due to the stepwise exclusion of interfering factors from the residuals. The contributions of different variables were compared using standardized regression coefficients, as shown in Figure 7. The results reveal that these factors exhibit distinct cyclical patterns. To account for these dynamics, we constructed training objectives based on the more challenging CAPM model. Although this approach increases the optimization difficulty, it ensures stable performance even when certain factors become less effective.

| Model | RAG | | | Zero-Shot | | | ICL | | | FinRipple/w-o alignment | | | FinRipple | | |
|---|---|---|---|---|---|---|---|---|---|---|---|---|---|---|---|
| | Coef. | p-value | $R^2$ | Coef. | p-value | $R^2$ | Coef. | p-value | $R^2$ | Coef. | p-value | $R^2$ | Coef. | p-value | $R^2$ |
| llama2-7b-chat | 0.021 | 0.482 | 0.013 | 0.040 | 0.657 | 0.021 | 0.058 | 0.287 | 0.145 | 0.090 | 0.520 | 0.152 | 0.310* | 0.021 | 0.275 |
| llama2-13b-chat | 0.132 | 0.405 | 0.074 | 0.095 | 0.445 | 0.065 | 0.158 | 0.245 | 0.138 | 0.182 | 0.314 | 0.195 | 0.445* | 0.013 | 0.390 |
| llama3-8b-instruct | 0.102 | 0.365 | 0.051 | 0.067 | 0.380 | 0.030 | 0.088 | 0.370 | 0.099 | 0.211 | 0.402 | 0.178 | 0.370 | 0.007 | 0.400 |
| vicuna-7b-chat | 0.158 | 0.235 | 0.095 | 0.112 | 0.400 | 0.078 | 0.215 | 0.142 | 0.134 | 0.250 | 0.188 | 0.256 | 0.515*** | 0.001 | 0.485 |
| vicuna-13b-chat | 0.505** | 0.028* | 0.145 | 0.172 | 0.210 | 0.123 | 0.290* | 0.031 | 0.255 | 0.365 | 0.175 | 0.342 | 0.610*** | 0.001 | 0.550 |
| Phi-3.5-mini-instruct | 0.097 | 0.512 | 0.032 | 0.056 | 0.670 | 0.026 | 0.075 | 0.470 | 0.086 | 0.153 | 0.395 | 0.202 | 0.285** | 0.005 | 0.335 |
| gemma-2-9b-it | 0.112 | 0.298 | 0.061 | 0.089 | 0.423 | 0.047 | 0.178 | 0.285 | 0.144 | 0.265 | 0.305 | 0.330 | 0.395*** | 0.001 | 0.445 |
| GPT 3.5 | 0.060 | 0.455 | 0.018 | 0.045 | 0.550 | 0.039 | 0.069* | 0.018 | 0.106 | / | / | / | / | / | / |
| GPT 4.0-preview | 0.165 | 0.328 | 0.045 | 0.119 | 0.389 | 0.063 | 0.195 | 0.512 | 0.138 | / | / | / | / | / | / |
| GPT 4o-mini | 0.198 | 0.215 | 0.051 | 0.145 | 0.312 | 0.055 | 0.155 | 0.209 | 0.121 | / | / | / | / | / | / |

Table 9: Differences in the explanatory power of Fama3 residuals by baselines and **FinRipple** applied to LLMs. Significance levels: * $p < 0.05$, ** $p < 0.01$, *** $p < 0.001$. Cells with '/' indicate unavailable model parameters.

| Model | RAG | | | Zero-Shot | | | ICL | | | FinRipple/w-o alignment | | | FinRipple | | |
|---|---|---|---|---|---|---|---|---|---|---|---|---|---|---|---|
| | Coef. | p-value | $R^2$ | Coef. | p-value | $R^2$ | Coef. | p-value | $R^2$ | Coef. | p-value | $R^2$ | Coef. | p-value | $R^2$ |
| llama2-7b-chat | 0.018 | 0.489 | 0.014 | 0.042 | 0.670 | 0.025 | 0.078 | 0.260 | 0.152 | 0.127 | 0.445 | 0.185 | 0.345** | 0.007 | 0.300 |
| llama2-13b-chat | 0.155* | 0.039 | 0.082 | 0.091 | 0.435 | 0.068 | 0.180 | 0.428 | 0.150 | 0.225 | 0.309 | 0.220 | 0.500*** | 0.001 | 0.420 |
| llama3-8b-instruct | 0.112 | 0.368 | 0.059 | 0.075 | 0.385 | 0.034 | 0.103 | 0.330 | 0.109 | 0.265 | 0.306 | 0.205 | 0.405*** | 0.001 | 0.440 |
| vicuna-7b-chat | 0.170* | 0.021 | 0.101 | 0.125 | 0.370 | 0.087 | 0.250 | 0.303 | 0.145 | 0.288 | 0.107 | 0.280 | 0.565*** | 0.001 | 0.525 |
| vicuna-13b-chat | 0.540** | 0.010 | 0.160 | 0.190* | 0.042 | 0.148 | 0.320 | 0.315 | 0.260 | 0.420 | 0.111 | 0.375 | 0.655*** | 0.000 | 0.590 |
| Phi-3.5-mini-instruct | 0.105 | 0.495 | 0.038 | 0.050 | 0.690 | 0.032 | 0.090 | 0.460 | 0.095 | 0.185 | 0.422 | 0.230 | 0.330** | 0.004 | 0.360 |
| gemma-2-9b-it | 0.140* | 0.028 | 0.068 | 0.087 | 0.425 | 0.048 | 0.205 | 0.727 | 0.155 | 0.305 | 0.267 | 0.360 | 0.430*** | 0.001 | 0.485 |
| GPT 3.5 | 0.070 | 0.435 | 0.023 | 0.038 | 0.585 | 0.039 | 0.085 | 0.322 | 0.120 | / | / | / | / | / | / |
| GPT 4.0-preview | 0.180* | 0.031 | 0.050 | 0.125 | 0.390 | 0.062 | 0.220 | 0.606 | 0.150 | / | / | / | / | / | / |
| GPT 4o-mini | 0.205 | 0.629 | 0.058 | 0.145 | 0.315 | 0.061 | 0.175 | 0.703 | 0.135 | / | / | / | / | / | / |

Table 10: Differences in the explanatory power of Fama3 residuals by baselines and **FinRipple** applied to LLMs. Significance levels: * $p < 0.05$, ** $p < 0.01$, *** $p < 0.001$. Cells with '/' indicate unavailable model parameters.

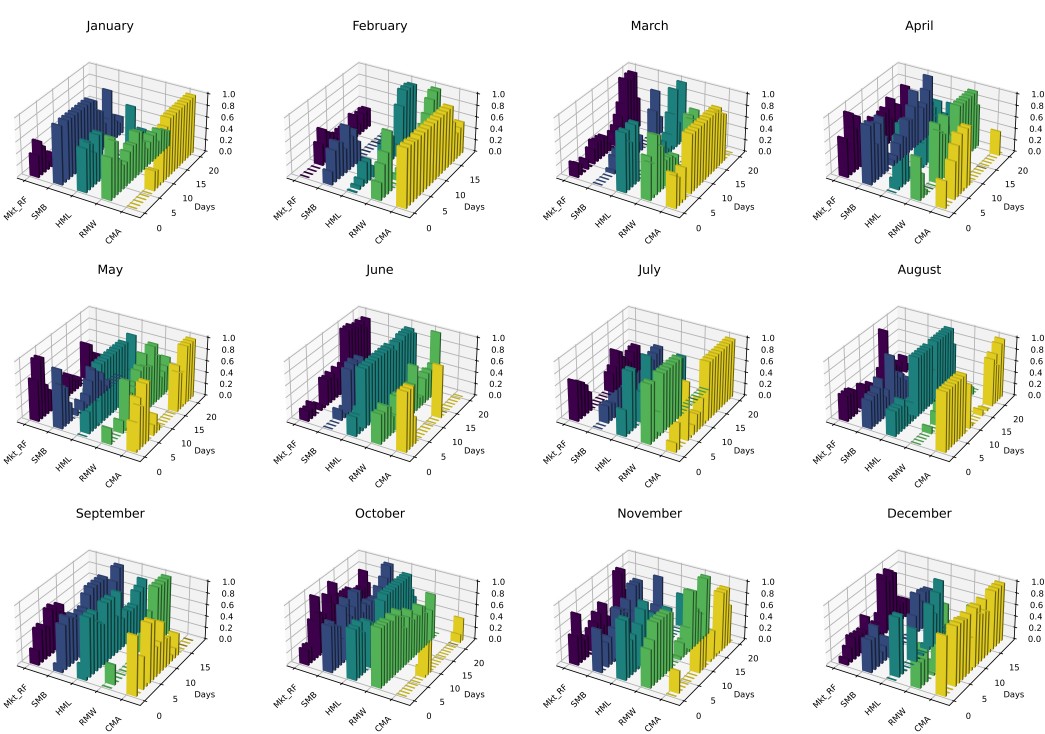

Figure 7: Variable importance of Fama-French 5 factors on 2018 returns.

# E BASELINES DETAILS

## E.1 ASSET PRICING

### E.1.1 ZERO SHOT

Zero-shot inference enables the model to analyze a wider range of market scenarios without relying on specific examples. The prompt used is shown as following:

```
Instruction:

You are a financial event analyst focused on analyzing the potential
impacts of news reports on the market. Based on the given news content
and current market structure, evaluate and output the affected companies
(TICKER in SP500), the type of impact (positive, negative, or neutral),
and a score representing the strength of the impact (ranging from -10
to +10, where -10 indicates a very negative impact and +10 indicates
a very positive impact). Provide specific company names and event
descriptions for clarity and utility. A market news report, company's
knowledge graph information, specific requirements and output format
will be provided below.

Market news report:

[INSERT MARKET NEWS REPORT]

Knowledge Graph (current market struture you can refer to):

[INSERT KNOWLEDGE GRAPH]

Requirement:

"Provide your result, strictly following the output format below,
without any additional output."

Output Format:

"Please provide your response in a structured JSON format. The JSON
should have a top-level object with a single key 'impact_analysis'.
The value of 'impact_analysis' should be an object containing two keys:
'affected_companies': An array of objects:
'name': The company's name (string)
'impact_type': The type of impact, e.g. 'positive' or 'negative' (string)
'impact_score': A numerical score representing the impact (integer)
'analysis': A string containing a brief analysis of the overall impact.
Please ensure that the JSON is properly formatted and uses double
quotes for strings.

Here's an example of how the structure should look:
{
    'impact_analysis': {
    'affected_companies': [
        {
        'name': 'Company Name',
        'impact_type': 'impact type',
        'impact_score': score
        },
        ...
    ],
    'analysis': 'Your analysis text here.'
    }
}"
```

### E.1.2 RAG AND ICL

To effectively analyze financial events and their market impact, we employ a ICL baseline. This method provides the model with a concrete example, demonstrating the expected input format, analysis process, and output structure. By presenting a sample scenario and its corresponding analysis, we establish a clear framework for the model to follow. For the RAG method, we use text-embedding-ada-002 as our embedding model, with the same prompt template as used in ICL. The following prompt illustrates this few-shot learning technique:

```
Instruction:

You are a financial event analyst focused on analyzing the potential
impacts of news reports on the market. Based on the given news content
and current market structure, evaluate and output the affected companies
(TICKER in SP500), the type of impact (positive, negative, or neutral),
and a score representing the strength of the impact (ranging from -10 to
+10, where -10 indicates a very negative impact, and +10 indicates a very
positive impact). Provide pecific company names and event descriptions
for clarity and utility. Here is an example.

Input Example:

 "Company A announces a partnership with Company B to jointly develop new
 technology, expected to significantly enhance production efficiency
 and increase market share."

Output Format Example:

{
  "impact_analysis": {
    "affected_companies": [
      {
        "name": "Company A",
        "impact_type": "positive",
        "impact_score": 8
      },
      {
        "name": "Company B",
        "impact_type": "positive",
        "impact_score": 6
      }
    ],
    "analysis": "The partnership between Company A and Company B is
    expected to enhance their technological capabilities and market
    competitiveness, likely increasing their revenues and stock prices.
  }
}

Input (you need to analyze):

 "Company A announces a partnership with Company B to jointly develop new
 technology, expected to significantly enhance production efficiency
 and increase market share."
Knowledge Graph (current market struture you can refer to):

(Company A, Company B, supplier)
(Company C, Company D, subsidiary)
(Company E, Company F, competitor)
(Company G, Company H, partner)
(Company I, Company J, investor)
(Company Q, Company R, technology provider) ...

Provide your result, strictly following the output format in the
example, without any additional output.
```

## E.2 STATISTICAL METRICS

This subsection introduces key statistical metrics used to evaluate the explanatory power of independent variables on the dependent variable, including Coefficient (Coef.), p-value, Coefficient of Determination ($R^2$), ANOVA F-statistic (ANOVA-F), ANOVA p-value (ANOVA-p), and Effect Size ($\eta^2$).

**Coefficient (Coef.)** The coefficient ($\beta_i$) represents the estimated effect of an independent variable $X_i$ on the dependent variable $Y$, holding all other variables constant. The regression equation is given by $Y = \beta_0 + \beta_1 X_1 + \beta_2 X_2 + \cdots + \beta_n X_n + \epsilon$, where $\epsilon$ is the error term.

**p-value** The p-value indicates the statistical significance of each coefficient, measuring the probability of observing the estimated effect under the null hypothesis that the coefficient is zero. A smaller p-value suggests stronger evidence against the null hypothesis.

**Coefficient of Determination ($R^2$)** The Coefficient of Determination ($R^2$) measures the proportion of variance in the dependent variable that is explained by the independent variables. It is calculated as $R^2 = 1 - \frac{\sum_{i=1}^{n}(y_i - \hat{y}_i)^2}{\sum_{i=1}^{n}(y_i - \bar{y})^2}$, where $y_i$ is the observed value, $\hat{y}_i$ is the predicted value, and $\bar{y}$ is the mean of the observed values.

**ANOVA F-statistic (ANOVA-F)** The ANOVA F-statistic tests whether the regression model explains a significant proportion of variance in the dependent variable compared to a model with no predictors. It is calculated as $F = \frac{\text{MS}_{\text{regression}}}{\text{MS}_{\text{residual}}}$, where $\text{MS}_{\text{regression}}$ is the mean square due to regression, and $\text{MS}_{\text{residual}}$ is the mean square due to residual error. Higher values of $F$ suggest a better fit of the model.

**ANOVA p-value (ANOVA-p)** The ANOVA p-value indicates the statistical significance of the F-statistic, reflecting the probability of obtaining the computed F-statistic under the null hypothesis that the regression model has no explanatory power.

**Effect Size ($\eta^2$)** Effect Size ($\eta^2$) represents the proportion of the total variance in the dependent variable that is attributable to an independent variable or a set of independent variables. It is calculated as $\eta^2 = \frac{\text{SS}_{\text{between}}}{\text{SS}_{\text{total}}}$, where $\text{SS}_{\text{between}}$ is the sum of squares between groups, and $\text{SS}_{\text{total}}$ is the total sum of squares. This metric helps determine the magnitude of the effect of the independent variables.

## E.3 PORTFOLIO MANAGEMENT

Portfolio management involves the selection and optimization of asset allocation to maximize the return within a given investment process (Hu and Lin, 2019). In this section, we describe the implementation details of five benchmark portfolio strategies: Equal Weighting, Volatility Weighting, Markowitz Model, Min-Variance Weighting, and **FinRipple**. These benchmarks are evaluated using metrics such as Daily Return ($R_d$), Sharpe Ratio ($S_a$), Maximum Drawdown (MDD), and Win Rate. In our experiments, we use historical data from the past 30 days as input. To simplify the comparison and ensure fairness, tax rates are set to zero across all scenarios.

### E.3.1 EQUAL WEIGHTING

The Equal Weighting strategy assigns an equal weight to each asset in the portfolio:

$$w_i = \frac{1}{N}, \quad i = 1, 2, \ldots, N \tag{9}$$

where $w_i$ represents the weight of asset $i$, and $N$ is the total number of assets.

### E.3.2 VOLATILITY WEIGHTING

The Volatility Weighting strategy allocates weights inversely proportional to the historical volatility of each asset:

$$w_i = \frac{\frac{1}{\sigma_i}}{\sum_{j=1}^{N} \frac{1}{\sigma_j}}, \quad i = 1, 2, \ldots, N \tag{10}$$

where $\sigma_i$ is the historical volatility (standard deviation) of asset $i$.

### E.3.3 MARKOWITZ MODEL

The Markowitz Model, also known as the Mean-Variance Optimization Model, aims to maximize expected return for a given level of risk or minimize risk for a given expected return:

$$\max_{\mathbf{w}} \quad \mathbf{w}^T \mu - \frac{\lambda}{2} \mathbf{w}^T \mathbf{\Sigma} \mathbf{w} \tag{11}$$

$$\text{s.t.} \quad \mathbf{1}^T \mathbf{w} = 1, \quad \mathbf{w} \geq 0 \tag{12}$$

Where $\mathbf{w}$ is the vector of portfolio weights, $\mu$ is the expected return vector, $\mathbf{\Sigma}$ is the covariance matrix of asset returns, and $\lambda = 1$ is the risk aversion parameter, representing a moderate balance between risk and return.

### E.3.4 MIN-VARIANCE WEIGHTING

The Min-Variance Weighting strategy seeks to construct a portfolio with the lowest overall risk:

$$\min_{\mathbf{w}} \quad \mathbf{w}^T \mathbf{\Sigma} \mathbf{w} \tag{13}$$

$$\text{s.t.} \quad \mathbf{1}^T \mathbf{w} = 1, \quad \mathbf{w} \geq 0 \tag{14}$$

where $\mathbf{\Sigma}$ is the covariance matrix of asset returns.

### E.4 METRICS OF PORTOFOLIO MANAGEMENT

The benchmarks are evaluated using the following metrics:

**Daily Return** ($R_d$)  The daily return measures the return of an asset over one day, calculated as $R_d = \frac{P_t - P_{t-1}}{P_{t-1}}$, where $P_t$ is the asset price at time $t$, and $P_{t-1}$ is the price on the previous trading day.

**Sharpe Ratio** ($S_a$)  The Sharpe ratio measures investment performance compared to a risk-free asset, adjusted for risk, using the formula $S_a = \frac{\bar{R}_a - R_f}{\sigma_a}$, where $\bar{R}_a$ is the average annual return, $R_f$ is the risk-free rate, and $\sigma_a$ is the standard deviation of the return.

**Maximum Drawdown (MDD)**  Maximum Drawdown represents the maximum observed loss from a peak to a trough of an asset's price, given by $\text{MDD} = \max_{t \in [1,T]} \left( \frac{\max_{j \in [1,t]} P_j - P_t}{\max_{j \in [1,t]} P_j} \right)$, where $P_t$ is the price at time $t$, and $T$ is the total time period considered.

**Win Rate (Wr)**  Win Rate represents the percentage of time periods in which the portfolio achieves a positive return, defined as $\text{Wr} = \frac{\sum_{t=1}^{T} \mathbb{I}(R_t > 0)}{T} \times 100\%$, where $R_t$ is the return at time $t$, $T$ is the total number of time periods considered, and $\mathbb{I}(R_t > 0)$ is an indicator function that equals 1 if $R_t > 0$, and 0 otherwise.

# F REPRODUCIBILITY STATEMENT

## F.1 HYPERPARAMETER SELECTION

We conducted hyperparameter tuning on a small-scale dataset to determine the optimal settings for minimizing the refusal-to-answer rate. The resulting hyperparameter settings are shown in Table 11, aiming to reduce the likelihood of model refusal while maintaining high response quality. In the reward function, $\lambda$ is set to 0.1. We used LoRA (Low-Rank Adaptation) (Hu et al., 2021) to fine-tune the model, with key settings including lora_alpha $= 16$, lora_dropout $= 0.1$, and rank $r = 64$.

| Model | Temperature | Top-k | Top-p |
|---|---|---|---|
| llama2-7b-chat | 0.8 | 40 | 0.85 |
| llama2-13b-chat | 0.7 | 50 | 0.90 |
| llama3-8b-instruct | 0.7 | 30 | 0.80 |
| vicuna-7b-chat | 0.8 | 45 | 0.88 |
| vicuna-13b-chat | 0.7 | 50 | 0.92 |
| Phi-3.5-mini-instruct | 0.9 | 35 | 0.86 |
| gemma-2-9b-it | 0.9 | 25 | 0.83 |
| GPT 3.5 | 0.8 | 30 | 0.80 |
| GPT 4.0-preview | 0.8 | 40 | 0.85 |
| GPT 4o-mini | 0.7 | 40 | 0.87 |

Table 11: Hyperparameter settings.

## F.2 COMPUTATIONAL RESOURCES AND CODE AVAILABILITY

The training and inference results required a total of over 9000 GPU hours using 25 A800 (80G) GPUs. We will release a user-friendly training framework along with the complete benchmark dataset in the future.

