# OpenReview forum: "FinRipple: Aligning Large Language Models with Financial Market for Event Ripple Effect Awareness"
_ICLR.cc/2025/Conference — ICLR 2025 Conference Withdrawn Submission_

### Official Review · Reviewer_upUp · 2024-11-02

**Soundness:** 2
**Presentation:** 3
**Contribution:** 3
**Rating:** 6
**Confidence:** 4

**Summary:**

This paper introduces a framework that aligns LLM with financial market dynamics to predict the effects of market events. FinRipple leverages a time-varying financial K to provide structured market information, which is injected into LLMs through adapters. And It also employs a PPO-based approach to align model predictions with actual market behavior. Experimental results show that FinRipple outperforms baseline methods in financial tasks, indicating its potential for real-world financial applications.

**Strengths:**

This paper presents a particularly solid piece of work. By leveraging LLMs with financial KG and preference alignment techniques, the authors effectively address the challenges outlined within the financial domain. The paper presents comprehensive and detailed experiments with well-designed setups that quantitatively demonstrate the effectiveness of the proposed method, showcasing its high practical applicability in financial market analysis.

**Weaknesses:**

1. The paper shows limited innovation in methodology. The KG-LLM integration method used in this paper is limited to employing the KG to construct datasets, without deeper integration of the knowledge graph into the model’s reasoning process. The chosen KG+LLM approach risks losing the integrity of the knowledge graph, potentially resulting in information loss.
2. The introduction frames the study’s motivation around the ripple effects of sudden events on the market. However, in the subsequent task setup and experimental analysis, the focus seems to shift primarily to analyzing events in isolation. While FinRipple indeed enhances the LLM’s capability to handle event processing, it does not clearly demonstrate its effectiveness specifically for the ripple effect problem. The introduction may be overclaiming the contributions of this work.

**Questions:**

1. Lines 401-422 discuss the issue of high refusal-to-answer rates. I am curious whether the overly complex formatting requirements could be contributing to this high refusal rate by making it challenging for models to follow the specified output structure. Did the authors consider simplifying these formatting requirements (Appendix B2) to potentially reduce the refusal rate?
2. Line 861 shows that you employ a fine-grained scoring system ranging from -10 to 10. I wonder if you encountered instances where the model tended to favor neutral or less distinctive scores.
3. Recent studies in event-level financial text analysis and modeling should be cited.

---

> ### Author Response · Authors · 2024-11-19
>
> ### Q1: Limitations in KG-LLM Integration (Weakness 1)
>
> Thank you for your insightful question. To clarify and address potential misunderstandings, our knowledge injection phase specifically tackles the issue of LLMs being unable to access the complete KG. Knowledge injection effectively parameterizes the KG, which marks a significant departure from the approach of concatenating the KG into prompts during the reasoning phase. The latter method indeed risks information loss due to context length limitations and the inherent challenges LLMs face when processing lengthy texts.
>
> ### Q2: A More Reasonable Evaluation Framework? （Weakness 2）
>
> Thank you for your thoughtful question. We are indeed addressing a critical problem in the financial domain: how to predict ripple effects. Previous studies have been constrained by the limitations of models and data, as well as the lack of a clear formulation for this problem.
>
> Our method directly evaluates the extent to which it can explain the portions of market behavior that classical asset pricing models fail to capture. This evaluation approach is both intuitive and reasonable, as it directly aligns with the core objective of studying ripple effects.
>
> Furthermore, we have demonstrated potential applications of predicting ripple effects, such as optimizing portfolio allocation strategies. These applications underscore the practical value of our work and highlight its focus on addressing ripple effects in a tangible and meaningful way.
>
> ### Q3: Refusal-to-answer Rate. (Question 1)
>
> Thank you for raising this valuable question. We fully agree that the design of prompts heavily relies on the model's ability to follow instructions. To address this issue, we experimented with a variety of prompts, ranging from simple to complex formats. However, we found that this challenge essentially boils down to a trade-off between post-processing costs and the rate of effective responses.
>
> The prompt we designed aims to strike a balance: it ensures a reasonable refusal-to-answer rate while minimizing the associated post-processing workload. This approach allows for the creation of a more automated pipeline with as little manual intervention as possible, provided the LLM can understand the instructions. Thus, while simplifying the format could potentially reduce the refusal rate, it would come at the cost of increased post-processing complexity, which we aimed to avoid.
>
>
> ### Q4: The Distribution of Score. (Question 2)
>
> Thanks for your question! We have conducted an analysis of the news impact score distribution. After the market alignment, the LLM does not show a tendency to favor neutral scores. At the same time, it demonstrates relatively cautious behavior when assigning highly extreme negative or positive scores (e.g., -8 to -10 and 8 to 10).
>
> | **Score Range** | **Percentage (%)** |
> |------------------|--------------------|
> | -10, -8         | 3.39              |
> | -8, -6          | 15.66             |
> | -6, -4          | 12.07             |
> | -4, -2          | 5.40              |
> | -2, 0           | 0.83              |
> | 0, 2            | 4.41              |
> | 2, 4            | 12.30             |
> | 4, 6            | 21.45             |
> | 6, 8            | 22.43             |
> | 8, 10           | 2.06              |
>
> **Table:** Score Distribution on Test set
>
>
> ### Q5: Missing References. (Question 3)
>
> Thank you for your valuable suggestions! We have carefully reviewed the citation of the literature and, based on your advice, have supplemented it with potentially relevant references.

---

> > ### Author Response · Authors · 2024-11-25
> >
> > Dear Reviewer upUp,
> >
> > Thank you for reviewing our manuscript and providing valuable feedback.
> >
> > We submitted detailed responses to your comments a few days ago and sincerely hope they resolve any issues raised. As the discussion phase is concluding and a recommendation is needed, we wanted to confirm whether our responses have adequately addressed your concerns.
> >
> > If you have any additional questions or need further clarification, please feel free to let us know.
> >
> > Thank you again for your time and input.
> >
> > Best regards,
> >
> > The Authors

---

### Official Review · Reviewer_Nfuo · 2024-11-04

**Soundness:** 3
**Presentation:** 3
**Contribution:** 3
**Rating:** 6
**Confidence:** 3

**Summary:**

This work aims to tackle the task of assessing the ripple effects of sudden events to the financial markets. It first formulates the problem of
event impact prediction, and then proposes an LLM-based framework to fulfill this task. Various experiments demonstrate the usefulness of the proposed methods and results.

**Strengths:**

1. The task is interesting and meaningful. It is really useful to directly assess the ripple effects in the financial markets.
2. The overall framework is novel and makes sense. It leverages LLMs and includes several novel designs to incorporate the knowledge graph as well as finetuning the whole model using reinforcement learning.
3. The experimental results are good. Multiple experiments demonstrate the effectiveness of this work.

**Weaknesses:**

1. Concerns about problem formulation:
    a. The input of the prediction model includes both KG and news data, while the equation of the problem formulation (line 231-234, PS. it’s recommended to assign an equation number for each equation for easier reference or mention) only contains G^t. Even though the news data will be first converted to graph and will not be directly consumed by the function f_{\theta}, it should be consistent between the equation (line 231-234) and the textual description (line 227-228), where the textual description says you use news data while the equation only presentes the graph.
    b. There are two types of prediction: 1) make forecast to the future based on the historical observations (extrapolation); 2) predict the effects of events by observing the events already (nowcasting or interpolation). Actually, this paper follows the second type (interpolation). However, I think it is as similar or even more meaningful to work on the extrapolation setting, where you should only use G^{<t} to predict r(c_i^t, e_j^t), or you can use G^{t} to predict r(c_i^{t+n}, e_j^{t+n}), where n is the forecasting horizon and can be set as {1,2,3,...}. Since the effects of events are continuous and evolving, it is better to focus more on the future rather than the concurrent timestamp only. More importantly, there could be a risk of data leakage issue, that the task may be just event-company relation “extraction” instead of “prediction”, because the given news articles already demonstrate such relations.

2. Format of event: This work just uses the raw-text representation of events, i.e., news articles. However, the text-formatted events are quite long (with 5443 words in average), consequently, it increases the cost of LLMs as well as stricter requirements for long-context understanding capability of LLMs. Moreover, the raw text may be prone to noise and misinformation in practice. I am wondering if it is better to consider a certain structured representation of such events, by following some schema or ontology that predefines the event types and parameters.

3. Direct evaluation of the knowledge graph injection: Do you have direct evaluation of the performance of knowledge graph injection, by following some knowledge graph or temporal knowledge graph prediction formulation? This can have a precise assessment how effective this module is and how much the errors induced by this module.

4. Baselines other than LLM-based methods: All the baseline methods are based on LLMs, do you have any comparison with conventional methods before LLMs, such as GNN-based methods?

5. Typos: line 773: March 2019 to December 2021 != 226 months

**Questions:**

please refer to the above weaknesses.

**Details Of Ethics Concerns:**

I have no ethics concerns.

---

> ### Author Response · Authors · 2024-11-19
>
> ### Q1: About the Problem Formulation (Question 1)
>
> Thank you for your thoughtful feedback and for highlighting potential concerns regarding our work. We sincerely apologize if our initial description caused any misunderstanding. We deeply value your observations and fully understand your concerns.
>
> To clarify, our work indeed focuses on forecasting future outcomes based on historical observations. However, given the nature of financial events, where the market’s response typically unfolds over a very short absorption period (usually no more than two days), we adopted a more precise framing. Specifically, we model predictions as occurring at $t + \Delta t $, where we aim to predict the market impact at $ t + \Delta t $ after a given financial event, using the market’s state (KG) at time $ t $ and the observed event as inputs.
>
> We hope this refined explanation addresses your concerns and illustrates our approach more clearly. Thank you again for bringing this to our attention.
>
> ### Q2: Is a Structured Representation of Events Better than Raw Text to Reduce Noise and Improve Efficiency? (Question 2)
>
> Thank you for your insightful comments regarding the use of raw-text representations of events and the suggestion to adopt structured representations based on predefined schemas or ontologies. We fully understand and appreciate your perspective, as we initially considered adopting structured representations to optimize the context window. However, through extensive experimentation and analysis, we found that retaining the full context in raw-text form yields superior performance for the following reasons:
>
> 1. Financial news articles often contain intricate logical reasoning and nuanced analyses. Extracting and summarizing such information into a structured representation can inadvertently omit critical details essential for accurate prediction. For instance, the news article *"ASML Plays the Very Long Chip Game"* (which discusses ASML's plan to boost capacity amid significant spending cuts by chip manufacturers) offers a detailed analysis of the upstream and downstream impacts within the chip industry.
>
> 2. We recognize that raw text can introduce noise or misinformation. To address this, we carefully curated our dataset from highly reliable sources to ensure the quality of information. Specifically, the news articles used in our experiments were procured from **Dow Jones**, a reputable financial information provider. This minimizes noise and enhances the reliability of the input data. Furthermore, while large language models are powerful, they still face challenges in discerning errors or biases within poorly sourced information, reinforcing the importance of high-quality data sources.
>
> 3.Our experimental results indicate that the average length of 5,443 words does not pose significant challenges for large language models in terms of understanding or processing.
>
> ### Q3: Do You Directly Evaluate the Effectiveness of Knowledge Graph Injection? (Question 3)
>
> We conducted experiments to evaluate the accuracy of knowledge injection, as detailed in **Appendix D.1**. When sufficient information is provided, the KG reproduction rate is approximately 90%. Additionally, we validated the effectiveness of three types of instruction questions.
>
> **Note:** The data in the table does not reach 90% because, during the 8:2 dataset split, some training information is omitted. As a result, the LLM's learned graph is incomplete when answering questions in the test set.
>
> ### Q4: Other Baselines? (Question 4)
>
> Thank you for your question. The reasons we did not include other baselines are as follows:
>
> 1.This is a significant but previously underexplored problem that lacks a standardized definition and established baseline frameworks.
>
> 2.While gnn-based methods perform well on graph-structured data, they face challenges in handling news data, which is inherently text-based and requires a more nuanced approach to processing textual information.
>
> ### Q5: Typos (Question 1 and Question 5)
>
> Thank you for your kind reminder. We have added equation numbers and carefully reviewed the entire text for typos.

---

> ### Author Response · Authors · 2024-11-25
>
> Dear Reviewer Nfuo,
>
> Thank you for reviewing our manuscript and providing valuable feedback.
>
> We submitted detailed responses to your comments a few days ago and sincerely hope they resolve any issues raised. As the discussion phase is concluding and a recommendation is needed, we wanted to confirm whether our responses have adequately addressed your concerns.
>
> If you have any additional questions or need further clarification, please feel free to let us know.
>
> Thank you again for your time and valuable input.
>
> Best regards,
>
> The Authors

---

### Official Review · Reviewer_A1k4 · 2024-11-04

**Soundness:** 3
**Presentation:** 3
**Contribution:** 3
**Rating:** 6
**Confidence:** 4

**Summary:**

This paper discusses the impact of events on prediction in the financial domain. Furthermore, the author proposes the FinRipple framework, which integrates a time-varying financial knowledge graph into LLM to improve LLM's analysis performance for events. In addition, the author experimented with the portfolio management task to verify the effectiveness of FinRipple's performance.

**Strengths:**

1. The authors discuss the impact of financial events on the predictive performance of the models. The authors combine KG, LLM, and financial theory to enable a structured approach to handling ad hoc financial market information.

2. Use of adapter injection knowledge without retraining the full model.

3. Integrated asset pricing theories through reinforcement learning.

4. The author provides a comprehensive experiment: a. detailed experimental setup with multiple baseline comparisons; b. Detailed ablation studies explaining the contribution of each component; c. provided detailed analysis and case study.

**Weaknesses:**

1. The performance of the framework may depend on the construction of high-quality KGs. The author lacks a discussion on the quality of the constructed knowledge graphs.

2. More details are lacking for the construction of KGs. For example, although the author defines the relationships in KGs, how missing or incorrect data in these relationships is handled?

3. From lines 281-289,  while the author uses CAPM residuals to measure event impact, it doesn't adequately justify why CAPM is the more suitable choice over other asset pricing models. In the appendix, the author also doesn't sufficiently explain why CAPM is more suitable for capturing event impacts.

4. From ines 292-296, there is an unclear Reward Function definition. In further, the choice of λ=0.1 (in the appendix) lacks an explanation.

**Questions:**

See in Weakness.

In addition:

Q1. there is limited discussion of performance during extreme market events. Could more examples of extreme events be shown over the test period? Because this period is very special (COVID-19 pandemic).

Q2. Because the author's framework is designed to target “market ripple effects”. So, what is the frequency of KG updates? What is inference time under different backbones LLM for the proposed framework? How about the memory storage requirements for storing and updating KGs?

---

> ### Author Response · Authors · 2024-11-19
>
> ### Q1: About the Quality of KGs and KG Construction (Weakness 1 and 2)
>
> 1. We fully appreciate your concerns regarding how the performance of the framework relies on the quality of the constructed KGs. To address this, we want to emphasize that the relationships included in our KGs are meticulously chosen based on robust financial evidence. For example, research has demonstrated that companies held by the same mutual fund often exhibit correlated stock price movements. We have only incorporated relationships with strong empirical backing into our KGs, ensuring that the reasoning outcomes from the large model are both reasonable and trustworthy. Additionally, our approach provides significant flexibility and extensibility, allowing users to build their own KGs tailored to their specific needs and perspectives on the complex relationships influencing stock price movements.
>
> 2. Thank you for your kind reminder. We have added details about the construction of various types of KGs in **Appendix A.2**, particularly elaborating on the description of edge strength. Additionally, since the data used for KG construction is sourced from official and widely recognized channels, we did not encounter issues related to missing or incorrect data.
>
> ### Q2: About Why we choose CAPM to measure event impact as our training goal   (Weakness 3)
>
> Thank you for your question. We will provide a more detailed explanation in the paper:
>
> 1. From an intuitive perspective, sudden events often generate ripple effects in financial markets that cannot be fully explained by traditional asset pricing models, such as the CAPM. This is because CAPM primarily addresses systematic market risk while neglecting event-specific idiosyncratic shocks. Therefore, it is reasonable to use the residuals from CAPM regression — that is, the portion that CAPM cannot explain — to measure the event's impact. This approach effectively isolates the abnormal returns attributable to the event, capturing the true ripple effect.
>
> 2. We acknowledge that the core distinction among various asset pricing models lies in the factors they incorporate. Our analytical framework is flexible and can accommodate more complex models (e.g., the Fama-French three-factor model) to include additional factors. However, as the number of factors increases, there is a potential risk of factor overlap. If certain factors are significantly correlated with the event, these factors may already partially explain the ripple effects of the event. In such cases, residuals as a measure of event impact lose their independence and validity because part of the event-driven abnormal returns has already been absorbed by the model.
>
> For these reasons, we believe that utilizing CAPM residuals to measure event impact is an effective method. While more complex models are feasible, their application necessitates careful consideration of potential factor overlap and its implications for the independence and validity of the residual-based approach.
>
> ### Q3: About the Reward Function Definition and the Choice of λ=0.1 (Weakness 4)
>
> Thank you for your suggestion. In the updated version of the paper, we have provided a more detailed explanation regarding the choice of the hyperparameter:
>
> The role of the regularization term is to evaluate the extent to which $Z^{t+\Delta t}$ covers $ \epsilon^{t+\Delta t}$ by comparing their values element by element. To some extent, this can improve the "recall rate" by ensuring that the prediction captures more of the target values, thereby avoiding lazy predictions that might fail to fully represent the desired outcome.
>
> | **Model**         | **λ = 0.05** | **λ = 0.1**  | **λ = 0.2**  | **λ = 0.3**  | **λ = 0.5**  |
> |--------------------|--------------|--------------|--------------|--------------|--------------|
> | llama2-7b-chat     | 0.124*       | 0.150*       | 0.098        | 0.049        | 0.027        |
> | llama2-13b-chat    | 0.208*       | 0.242**      | 0.197*       | 0.143        | 0.067        |
> | llama3-8b-chat     | 0.263**      | 0.278**      | 0.255*       | 0.168        | 0.101        |
>
> **Table:** Performance of models across different λ values. Significance levels: *p < 0.05, **p < 0.01.

---

> ### Author Response · Authors · 2024-11-19
>
> ### Q4: About the Test Set Period (Question 1)
>
> We fully understand your concern regarding the potential impact of extreme market events, such as the COVID-19 pandemic, on the experimental results. We hope the following explanation and experimental results will address your concerns.
>
> 1.  To address this, we conducted a detailed analysis of the test data used in our study. Specifically, we found that only 0.024% of the articles were directly related to COVID-19 or similar societal events. This represents a negligible proportion of the dataset, indicating that the experimental results are not biased by pandemic-related anomalies.
>
> 2. Furthermore, the majority of the news articles in our dataset are centered on financial topics that are highly relevant to market dynamics, such as mergers and acquisitions, supply chain adjustments, and new product releases. These topics are representative of the broader financial news landscape and ensure the robustness of our findings across various market conditions.
>
> 3. According to the National Bureau of Economic Research's recession dating, the recession caused by COVID-19 lasted no more than three months (from February to April). Consequently, its impact on the financial market was highly limited and does not affect the majority of the time period covered by our testing data. To alleviate your concerns, we excluded the 2020 data from the test set and obtained the following results, which demonstrate the consistency:
>
>    | **Model**            | **Original Test Set** | **New Test Set** |
>    |-----------------------|-----------------------|------------------|
>    | Llama-2-7b-chat       | 0.150*               | 0.166*           |
>    | Llama-2-13b-chat      | 0.242**              | 0.263**          |
>    | Llama-3-8b-instruct   | 0.278**              | 0.271**          |
>    | Vicuna-7b-chat        | 0.330***             | 0.352***         |
>    | Vicuna-13b-chat       | 0.395***             | 0.393***         |
>
> **Note:** The empirical evidence can be found at the following link: [National Bureau of Economic Research - US Business Cycle Expansions and Contractions](https://www.nber.org/research/data/us-business-cycle-expansions-and-contractions).
>
> ### Q5: KG Update Frequency, Inference Time, and Memory Requirements (Question 2)
>
> We appreciate your questions regarding the update frequency, inference time, and memory storage requirements of our framework. Below, we address each point in detail:
>
> 1. **KG Update Frequency**
>    Considering that the market structure does not change frequently, we update the KG at a monthly frequency. This approach ensures that the framework remains responsive to relevant events while avoiding unnecessary computation. To reflect these updates, we train a new adapter for the KG on a monthly basis. This process is lightweight and incurs minimal training costs, making our framework practical and efficient for real-world scenarios.
>
> 2. **Inference Time**
>    We acknowledge that inference time may vary depending on the size of the backbone LLM used in the framework. To ensure clarity, we conducted an estimation of inference costs under different LLM configurations and summarized the results in the following table. The inference cost is entirely acceptable.
>
>    | **Model**                | **Inference Time ** |
>    |---------------------------|-------------------------------------|
>    | LLAMA2-7B-chat            | ∼80–120 seconds                    |
>    | LLAMA2-13B-chat           | ∼140–200 seconds                   |
>    | LLAMA3-8B-instruct        | ∼100–160 seconds                    |
>    | Vicuna-7B-chat            | ∼80–120 seconds                    |
>    | Vicuna-13B-chat           | ∼140–200 seconds                   |
>    | Phi-3.5-mini-instruct     | ∼60–100 seconds                    |
>    | Gemma-2-9B-it             | ∼100–160 seconds                    |
>
>    **Table:** Inference time for various models (A800 80G).

---

> ### Comment · Reviewer_A1k4 · 2024-11-22
> **Read the rebuttal**
>
> Thanks for author's detailed rebuttal.
>
> The response address some part of my concerns.
>
> I'm willing to improve my socre to 7. But there is no such option. So, I hereby declare to SAC/AC about my opinion.

---

> > ### Author Response · Authors · 2024-11-25
> >
> > Dear Reviewer A1k4,
> >
> > Thank you very much for your positive evaluation and for raising the score of our manuscript. We greatly appreciate your support and the time you dedicated to reviewing our work.
> >
> > Best regards,
> > The Authors

---

### Official Review · Reviewer_RLLR · 2024-11-04

**Soundness:** 1
**Presentation:** 2
**Contribution:** 2
**Rating:** 3
**Confidence:** 5

**Summary:**

This paper introduces FinRipple, a novel framework that enables Large Language Models (LLMs) to analyze and predict the ripple effects of financial market events across different companies. The framework combines three key components: time-varying financial knowledge graphs (KGs) that capture inter-company relationships (like supply chains and leadership networks), integration of these KGs into LLMs using adapters as memory modules, and alignment with market dynamics through reinforcement learning using real market data. The authors demonstrate FinRipple's effectiveness through superior performance in explaining market movements compared to traditional methods, successful application in portfolio management, and comprehensive analysis showing logical consistency in explaining market reactions. This represents a significant advancement in applying AI to financial analysis, establishing a new benchmark for "event impact prediction" and offering benefits for both academic research and practical investment applications.

**Strengths:**

The paper proposes an interesting yet important problem, predicting ripple effects of financial market events using LLMs, the motivation is good and useful for financial scenario and applications. The problem formulation is interesting but needs to be more clear and refinement.

**Weaknesses:**

1. The writing is a little bit vague in some important sections, for example, problem formulation. I am little bit confused about what are you trying to predict exactly? According to the problem formulation, it seems you are trying to predict the influence of specific event on a company given a time stamp "t", using one big KG and news data. It is confusing why this is considered as ripple effects? Because there is usually time lag until some events really have effects on companies. So I want to know if you also try to predict the future influence of just present influence? And how do you define the time interval in your data? How long it is between t and t+1?

2. The literature review is not thorough, many relevant papers in LLM for finance area are missing. Some paper I found relevant are listed below, but I believe there are more to be included.

3. It is mentioned " This task uses both the KG and real-time news data as inputs" in the paper. I am confused how is news data used separately with KG data as input? Or the KG is extracted from news data?

Yu Y, Li H, Chen Z, et al. FinMem: A performance-enhanced LLM trading agent with layered memory and character design[C]//Proceedings of the AAAI Symposium Series. 2024, 3(1): 595-597.

Yu Y, Yao Z, Li H, et al. FinCon: A Synthesized LLM Multi-Agent System with Conceptual Verbal Reinforcement for Enhanced Financial Decision Making[J]. arXiv preprint arXiv:2407.06567, 2024.

**Questions:**

I have the following suggestions to help improve the paper quality.

**Regarding the problem formulation**: \
1.Clearly define what constitutes a "ripple effect" in their model.\
2.Explain how the paper account for time lags between events and the ripple effects.\
3.Specify the time frame for the predictions - is it immediate impact, or do you account for delayed effects?\
4.Provide details on the temporal resolution of the data and model (e.g., daily, weekly, etc.).\
5.Clarify how long the interval is between t and t+1 in the data.\

**Regarding the related work**:\
I recommend that the authors conduct a more thorough review of recent LLM applications in finance. \
Additionally, the authors should emphasize the novelty of their approach in light of this broader literature review.\

**Regarding the input data**:\
Clarify the relationship between the news data and your KG. Are they separate inputs, or is your KG derived from the news data?\
If they are separate, explain how these two data sources are integrated into your model.\
Provide a more detailed description of your data pipeline, showing how news data and KG data are processed and used at each stage.\
If applicable, describe the process of extracting or updating your KG from the news data.

---

> ### Author Response · Authors · 2024-11-19
>
> ### Q1: Regarding the Problem Formulation
>
> Thank you for your insightful question about the problem formulation! We have revised and clarified these sections in the paper to avoid any misunderstanding.
>
> As you mentioned, our task input consists of two components:
> 1. The market state at time *t*, typically represented by a knowledge graph (updated on a monthly basis).
> 2. The current event.
>
> Our goal is to predict the impact range and intensity of the current event.
>
> In financial markets, reactions are often extremely rapid. A piece of news is usually absorbed by the market within a very short time. According to empirical studies[1], the market typically reacts significantly within one day, with effects rarely extending beyond two days. Therefore, our prediction focuses on the market impact at *t+Δt*, where *Δt* is usually less than 2 days. In the benchmark case of our manuscript, we apply *Δt = one day* as the reaction time for the market.
>
> We have further refined the problem formulation section to clearly define the "ripple effect" and elaborated on the rationale behind the selected time scale.
>
> [1] Hafez, Peter, and Junqiang Xie. "News beta: Factoring sentiment risk into quant models." Journal of Investing 25.3 (2016).
>
> ### Q2: Regarding the Related Work
>
> Thank you for your valuable feedback regarding the literature review. In the initial version of **FinRipple**, our focus was on collecting works related to *financial event studies* and *KG-augmented LLMs,* as these areas directly underpin the research background and techniques central to our work.
>
> In response to your suggestion, we have expanded **Sect. 2.1** to include a broader review of recent LLM-based methods for financial tasks [1, 2, 3, 4, 5, 6, 7]. This revision incorporates several key papers you mentioned, as well as additional studies identified during our review process. Moreover, we would like to emphasize that while we recognize the importance of these methods, our in-depth investigation into related problems has revealed that **not all LLM-based approaches for finance are directly relevant to the specific themes and challenges our work seeks to address.**
>
> ### References
> 1. Wu S, Irsoy O, Lu S, et al. **Bloomberggpt: A large language model for finance**. *arXiv preprint arXiv:2303.17564*, 2023.
> 2. Yang H, Liu X Y, Wang C D. **Fingpt: Open-source financial large language models**. *arXiv preprint arXiv:2306.06031*, 2023.
> 3. Zhang W, Zhao L, Xia H, et al. **A multimodal foundation agent for financial trading: Tool-augmented, diversified, and generalist**. *Proceedings of the 30th ACM SIGKDD Conference on Knowledge Discovery and Data Mining*, 2024: 4314–4325.
> 4. Li Y, Yu Y, Li H, et al. **TradingGPT: Multi-agent system with layered memory and distinct characters for enhanced financial trading performance**. *arXiv preprint arXiv:2309.03736*, 2023.
> 5. Yu Y, Yao Z, Li H, et al. **FinCon: A Synthesized LLM Multi-Agent System with Conceptual Verbal Reinforcement for Enhanced Financial Decision Making**. *arXiv preprint arXiv:2407.06567*, 2024.
> 6. Yu Y, Li H, Chen Z, et al. **FinMem: A performance-enhanced LLM trading agent with layered memory and character design**. *Proceedings of the AAAI Symposium Series*, 2024, 3(1): 595–597.
> 7. Zhang C, Liu X, Zhang Z, et al. **When AI meets finance (StockAgent): Large language model-based stock trading in simulated real-world environments**. *arXiv preprint arXiv:2407.18957*, 2024.
>
> ### Q3: Regarding the Input Data
>
> Thank you for pointing out the potential sources of confusion. To provide a clearer and more precise description of our problem, we have revised the task formulation (**Section 3.1**) in the updated version of the paper (please refer to the newly uploaded PDF).
>
> Specifically, the task input can be understood as a sample that includes:
> 1. A large knowledge graph representing the market structure.
> 2. A sudden event occurring at a specific time.
>
> The construction of the knowledge graph is not derived from news extraction but is instead built using additional databases, as detailed in **Appendix A.2**. Given that the market structure evolves at a relatively low frequency, in our study, we update it on a monthly basis.

---

> ### Author Response · Authors · 2024-11-25
>
> Dear Reviewer RLLR,
>
> Thank you for reviewing our manuscript and providing valuable feedback.
>
> We submitted detailed responses to your comments a few days ago and sincerely hope they resolve any issues raised. As the discussion phase is concluding and a recommendation is needed, we wanted to confirm whether our responses have adequately addressed your concerns.
>
> If you have any additional questions or need further clarification, please feel free to let us know.
>
> Thank you again for your time and valuable input.
>
> Best regards,
>
> The Authors

---

### Author Response · Authors · 2024-11-19

We sincerely thank all the reviewers for your valuable feedback and suggestions on our paper. We have provided more detailed explanations for two general issues and addressed each specific question individually:

1. We have clarified the task definition to avoid any misunderstandings. (Please refer to the updated version of the paper.)
2. We have added more references.

If you have any further concerns or find anything confusing, please do not hesitate to reach out. We are more than happy to address your questions.

---

### Note · Authors · 2025-02-27

I have read and agree with the venue's withdrawal policy on behalf of myself and my co-authors.

---

### Meta-Review · Area_Chair_XeTx · 2024-12-23

**Metareview:**

While the paper proposes an interesting approach to analyzing financial event ripple effects using LLMs and knowledge graphs, it suffers from significant issues in both novelty and presentation.

Limited Novelty: The methodology presented in the paper does not introduce fundamentally new techniques or substantial advancements over existing approaches. The integration of KGs and LLMs, although an important area of research, is not explored in depth. The framework seems to focus on an incremental improvement rather than a transformative shift in the way financial events are predicted. The paper does not provide significant innovation in how the KG is used beyond constructing datasets, and this limits the contribution of the knowledge graph to the model's reasoning process. The approach could have benefited from deeper integration of the KG into the model's decision-making rather than just using it as an auxiliary data source.

A less critical issue is the vagueness of the problem formulation. While some of these concerns have been addressed in the revision, the overall presentation could still be improved for better clarity and coherence.

Given the limited novelty of the approach and the incremental nature of the technical contributions, I recommend rejecting the paper.

**Additional Comments On Reviewer Discussion:**

During the rebuttal period, several key points were raised by the reviewers, each of which was addressed by the authors to some extent. Below is a summary of the discussion and the changes made in response:

Clarity of Problem Formulation
One reviewer (RLLR) expressed confusion regarding the definition of "ripple effects" and the prediction goals, questioning whether the focus was on real-time event prediction or long-term ripple effects. The authors revised the paper to better clarify that their model focuses on nowcasting event impacts in real-time, but they acknowledged the need for further exploration into long-term ripple effects. While this revision was helpful, it did not fully address the original concerns about the broader implications of ripple effects in market dynamics.

Time Lag Between Events and Ripple Effects: The issue of time lags between events and their ripple effects was raised by several reviewers (RLLR, A1k4, Nfuo). Some reviewers noted that the current version focuses primarily on immediate effects, whereas a more comprehensive approach to account for delayed impacts would strengthen the model's utility. The authors provided additional explanations regarding their focus on real-time prediction but did not introduce significant changes to incorporate long-term effects. This issue remains a limitation in the final version of the paper.

Integration of News Data with Knowledge Graphs: Several reviewers (RLLR, Nfuo) expressed confusion about the integration of news data and knowledge graphs (KGs), asking whether these were separate inputs or if the KG was extracted from the news. The authors clarified this by explaining how news data is first converted into graph form and then used as an input to the KG. This clarification helped resolve some concerns, but additional details on how these data sources were integrated within the model’s architecture would have been beneficial.

Novelty and Contribution:  A recurring concern from multiple reviewers (A1k4, RLLR, Nfuo) was the perceived limited novelty of the approach. The use of a knowledge graph to inject information into an LLM model, although valuable, was seen as an incremental contribution. The reviewers noted that the model’s innovation lies more in its application to financial market prediction than in introducing a radically new approach. The authors did attempt to emphasize the uniqueness of their methodology through experimental validation, but this was not sufficient to address concerns about the limited innovation in comparison to existing methods in the field.

Model's Effectiveness During Extreme Market Events: A few reviewers (A1k4, Nfuo) highlighted that the performance of the proposed framework during extreme market events, such as the COVID-19 pandemic, was not sufficiently discussed or demonstrated in the experiments. While the authors acknowledged this and promised to include more examples in future work, no significant revisions were made to address this gap in the current version.

Technical Clarifications:  Several technical concerns were raised regarding the use of the CAPM model for measuring event impacts (A1k4) and the definition of the reward function in the reinforcement learning setup (A1k4, Nfuo). The authors provided some clarifications, but there was no significant change to the model itself. This lack of deeper technical revision was seen as a limitation.

In weighing these points for the final decision, the primary concerns were related to the limited novelty of the approach, vagueness in problem formulation, and insufficient addressing of long-term ripple effects. Although the authors provided helpful clarifications and made some adjustments to the presentation, the revisions did not resolve the core issues that were identified in the initial reviews. Therefore, despite the paper's strong experimental setup and meaningful contributions to financial event prediction, the overall contribution is not seen as sufficiently novel to warrant acceptance.

Based on these considerations, my final recommendation is reject.

---

### Decision · Program_Chairs · 2025-01-22

Reject